# Effects of warm-season feeding on yak growth, antioxidant capacity, immune function, and fecal microbiota

Yining Xie,[1,2] Yangji Cidan,[1] Zhuoma Cisang,[1] Deji Gusang,[1] Quzha Danzeng,[1] Wangdui Basang,[1] Yanbin Zhu[1]

**ABSTRACT**  The yak (*Bos grunniens*) is of great importance to the local ecosystem and animal husbandry on the Tibetan Plateau. However, the impacts of different feeding practices on yak growth, health, and ecosystem interactions are not fully understood. This study investigates the effects of warm-season grazing and housing-feeding on yak growth performance, antioxidant capacity, immune function, metabolome, and fecal microbiota. The study found that grazing significantly increased the final body weight and average daily gain of yak ($P < 0.05$), reduced serum globulin and urea nitrogen levels, and elevated aspartate aminotransferase (AST) levels. Grazing enhanced serum total superoxide dismutase (T-SOD) and total antioxidant capacity (T-AOC). It also increased levels of immunoglobulins (IgA, IgM, IgG) and pro-inflammatory cytokines (IL-2, IL-6, TNF-α, IFN-γ). Meanwhile, grazing decreased levels of IL-4 and IL-10. Additionally, grazing significantly altered the plasma metabolite profile, particularly in bile acid metabolism pathways. The relative abundance of beneficial microbial genera (e.g., *Christensenellaceae_R-7_group*, *Monoglobus*, *Romboutsia*) in the feces of grazing yak was significantly higher, while total short-chain fatty acids were lower than in penned yak. Grazing improved growth performance and nutritional metabolism efficiency, enhanced antioxidant and immune functions, and optimized the structure of the gut microbiota in yak. These findings indicate that grazing can better utilize natural forage resources to promote yak health and improve production performance.

**IMPORTANCE**  This study investigates how different feeding patterns—grazing versus housing-feeding—affect the health, growth, and microbiome of yaks in the warm season. Yaks are vital to the Tibetan Plateau's ecosystem and local livelihoods. Understanding how feeding practices impact their health can help optimize yak management, ensuring better welfare and productivity. Grazing yaks showed improved growth, enhanced antioxidant and immune functions, and a healthier gut microbiota compared to penned yaks. These findings highlight the importance of natural forage in promoting yak health and could guide sustainable yak husbandry practices, benefiting both the animals and the communities that rely on them.

**KEYWORDS**  yak, feeding pattern, immune function, fecal microbiota

The yak (*Bos grunniens*), a unique ruminant native to the Tibetan Plateau, serves as the cornerstone of local animal husbandry by providing meat, milk, fiber, and essential ecosystem services (1, 2). As one of the few domesticated ungulates capable of thriving at altitudes above 3,000 m, yaks have evolved remarkable physiological and metabolic adaptations to withstand extreme environmental pressures, including hypoxia, diurnal temperature fluctuations, and seasonal forage shortages (3–5). However, yak husbandry systems are undergoing significant changes due to increasing human activities and the impacts of climate change on alpine meadows (6–8). The rapid development of modern animal husbandry has introduced new feeding practices, such as housing-feeding and

**Peer Reviewers** Lizhuang Hao, Qinghai University, Xining, China; Tariq Shah, Lanzhou University, Lanzhou, Gansu, China

Address correspondence to Yanbin Zhu, zhuyanbin126@126.com, or Wangdui Basang, bw0891@163.com.

Yining Xie and Yangji Cidan contributed equally to this article. Author order was determined based on their contributions to the study.

The authors declare no conflict of interest.

See the funding table on p. 12.

supplementary feeding (9–11). These practices are designed to optimize yak growth rates, improve feed utilization, and reduce environmental disturbances (12–14). Despite these advancements, grazing remains an extremely important production method in yak husbandry on the Tibetan Plateau. This system takes advantage of the seasonal abundance of grassland resources, especially during the warm season (May to September), when monsoon rains promote vigorous plant growth and increase the nutritional value of forage (15, 16). In traditional grazing, yaks roam freely over vast grasslands, exploiting natural forage resources, which are crucial for various aspects of their biology, including growth, health, and metabolism (6, 17, 18).

Despite the evident importance of grazing for yak and the ecosystem, the mechanisms by which warm-season grazing and housing-feeding affect yak's metabolic regulation, oxidative stress balance, immune capacity, hormone levels, gut microbiota dynamics, and overall health remain unclear. Filling these knowledge gaps is crucial for optimizing yak management practices in high-altitude regions. This study aims to elucidate the effects of warm-season grazing and housing-feeding on yak growth, physiology, antioxidant capacity, immune function, hormone levels, metabolome, and microbiome. We hypothesize that warm-season grazing improves metabolic efficiency and immune responses compared to housing-feeding by utilizing natural forage resources, reducing confinement stress, and optimizing the gut microbiota structure. In contrast, housing-feeding, while providing more stable nutrient intake, may increase stress and alter metabolic and microbial profiles. By comparing these two feeding systems, we hope to identify the optimal management strategies that improve production efficiency and animal welfare while maintaining ecological sustainability. Our findings will provide a scientific basis for sustainable yak husbandry on the plateau.

## MATERIALS AND METHODS

### Experimental design and management

The experiment was conducted from June to October 2024, during the warm season on the Tibetan Plateau. A total of 36 one-year-old yaks, half male and half female, and an initial average body weight of 56.08 ± 4.45 kg, were randomly assigned to two treatment groups: a grazing group and a housing-feeding group, with 18 yaks in each group. The entire experimental period lasted 105 days, including a 15-day pre-feeding period and a 90-day experimental period. The experiment was carried out in Linzhou, Lhasa, Tibet. The yak in the grazing group freely grazed on the natural pastures at an altitude of 3,335 m (29°45′–30°08′ N and 90°51′–91°28′ E) from 08:00 to 19:00 daily, and the barns for the stall-fed yak were also located in this area. The yak had free access to clean spring water and pasture forage, which mainly consisted of *Kobresia pygmaea*, *Festuca ovina*, *Carex moorcroftii*, *Stipa purpurea*, and others. The yaks in the housing group were housed in standard feeding pens (3 m × 3 m) equipped with stainless steel feed and water troughs, with one yak per pen ($n = 18$). They were provided with feed and water twice daily at 09:00 and 18:00, allowing free access to feed and water. The composition and nutritional level of the feed are shown in Table 1, and the daily feed intake of each pen was recorded. During the trial period, none of the yaks received any antibiotics, vaccines, or other treatment.

### Sample collection

On day 1 and day 90 of the experiment, all yaks were weighed after a 24 h fasting period ($n = 18$). On day 90, blood samples were collected from the tail root of the yak using vacuum blood collection tubes with or without sodium heparin. The samples were kept in the dark for 30 min and then centrifuged at 3,000 rpm for 15 min to separate plasma and serum. Fresh fecal samples were collected using the rectal massage method. All samples were initially preserved on dry ice and then transferred to a −80°C freezer for subsequent analysis.

**TABLE 1** Ingredients and nutritional levels of experimental diets

| Diet composition | Content (% of DM) | Nutritional level[a] | |
|---|---|---|---|
| Alfalfa hay | 15.00 | CP, % | 11.62 |
| Oat hay | 15.00 | EE, % | 4.06 |
| Full corn silage | 20.00 | NDF, % | 31.69 |
| Corn | 38.00 | ADF, % | 18.80 |
| Rapeseed oil | 0.50 | ADL, % | 3.56 |
| Wheat bran | 5.00 | NFC, % | 48.32 |
| Soybean meal | 1.50 | Starch, % | 28.2 |
| Cottonseed meal | 1.00 | Ash, % | 4.31 |
| Rapeseed meal | 1.00 | Ca, % | 0.86 |
| Calcium carbonate | 1.50 | P, % | 0.41 |
| Calcium hydrogen phosphate | 0.50 | ME, MJ/kg DM$^3$ | 7.82 |
| Premix[b] | 1.00 | | |

[a]The ME value of total mixed ration (TMR) was calculated based on the available ME data of the ingredients. CP, EE, NDF, ADF, ADL, Ca, and P were determined values. NFC = DM − (EE% + CP% + CP% + ASH%). DM, dry matter; ME, metabolizable energy; CP, crude protein; NDF, neutral detergent fiber; NFC, non-fiber carbohydrate; ADF, acid detergent fiber; ADL, acid detergent lignin; Ca, calcium; P, phosphorus.

[b]The premix provides per kg diet: 10 mg Cu in the form of sulfate, 60 mg Zn in the form of sulfate, 50 mg Mn in the form of sulfate, 50 mg Fe in the form of sulfate, Co in the form of chloride 0.2 mg, I in the form of iodate 0.5 mg, Se in the form of selenite 0.3 mg, and Vitamin A 10,000 IU, Vitamin D$_3$ 2,000 IU, and Vitamin E 60 IU.

## Determination of serum indicators

Serum total protein (TP) was measured using the Doumas method diagnostic kit, serum albumin (ALB) using the BCG method diagnostic kit, and serum urea nitrogen (BUN) using the BUN method diagnostic kit. The instructions provided with each kit were strictly followed to ensure accurate detection of TP, ALB, and BUN levels. Serum total cholesterol (TC), triglycerides (TG), and glucose (GLU) were measured using diagnostic kits based on the cholesterol oxidase-peroxidase (CHOD-POD) method for TC, the glycerol phosphatase oxidase–phenol4-amino antipyrene peroxidase (GPO-PAP) method for TG, and the modified glucose oxidase method for GLU, respectively. Serum alanine aminotransferase (ALT), aspartate aminotransferase (AST), and alkaline phosphatase (ALP) were measured using diagnostic kits based on the CC kinetic method for ALT, the IFCC kinetic method for AST, and a specialized ALP detection method, respectively.

Serum levels of total superoxide dismutase (T-SOD), total antioxidant capacity (T-AOC), malondialdehyde, and glutathione peroxidase (GSH-Px) were measured using diagnostic kits specific for each analyte. To accurately determine the levels of immune and growth-related indicators in yak serum, bovine-specific enzyme-linked immunosorbent assay kits were used to measure immunoglobulins IgA, IgM, and IgG, as well as cytokines such as interleukin-2 (IL-2), IL-4, IL-6, IL-10, tumor necrosis factor-α (TNF-α), interferon-γ (IFN-γ), growth hormone (GH), insulin-like growth factor-1 (IGF-1), growth hormone-releasing hormone (GHRH), and growth hormone-inhibiting hormone (GHIH).

## Plasma metabolomics

Metabolite extraction was performed by adding 50 mg of solid sample to a 2 mL centrifuge tube with a 6 mm grinding bead, followed by the addition of 400 μL extraction solution (methanol:water = 4:1, containing 0.02 mg/mL L-2-chlorophenylalanine as an internal standard). The samples were ground at −10℃ for 6 min and ultrasonicated at 5℃ for 30 min, then left at −20℃ for 30 min and centrifuged at 4℃ and 13,000 × $g$ for 15 min. The supernatant was transferred to an injection vial for liquid chromatography-tandem mass spectrometry (LC-MS/MS) analysis. A pooled quality control (QC) sample was prepared by mixing equal volumes of all samples and was injected regularly to monitor analytical stability. LC-MS/MS analysis was conducted using a Thermo UHPLC-Exploris 240 system with an ACQUITY HSS T3 column. The mobile phases consisted of 0.1% formic acid in water:acetonitrile (95:5) and 0.1% formic acid in acetonitrile:isopropanol:water (47.5:47.5:5). Gradient elution was performed with

different profiles for positive and negative ion modes, and MS conditions were optimized with an ESI source, operating at various temperatures, gas flow rates, and voltages. Data acquisition was performed in Data Dependent Acquisition mode over a mass range of 70–1,050 m/z.

## Fecal microbiota

In the study, we successfully extracted genomic DNA from yak feces using the E.Z.N.A. Soil DNA Kit (manufacturer: Omega Bio-tek, Norcross, GA, USA). The extracted DNA was then analyzed by 1% agarose gel electrophoresis. Subsequently, specific primers 338F (5′-ACTCCTACGGGAGGCAGCAG-3′) and 806R (5′-GGACTACHVGGGTWTCTAAT-3′) were employed to amplify the V3-V4 hypervariable region of the bacterial 16S rRNA gene. The resulting amplicons were sequenced on the Illumina MiSeq PE300 platform (manufacturer: Illumina, San Diego, USA) to obtain more precise microbiome data. To analyze the data, the research team used Uparse (version 7.0) to cluster the sequences at a 97% similarity level to identify operational taxonomic units (OTUs) and to remove chimeric sequences. Finally, the 16S rRNA database was classified using the RDP Classifier (version 2.11), with a classification threshold set at 0.7.

## Fecal short-chain fatty acids

For the experiment, short-chain fatty acid (SCFA) standard solutions were prepared by mixing eight SCFAs (acetic, propionic, butyric, isobutyric, valeric, isovaleric, hexanoic, and isohexanoic acids) in 9,840 µL of high-performance liquid chromatography-grade butanol to create stock solution A. An internal standard stock solution B was prepared by adding 10 µL of 2-ethylbutyric acid to 9,990 µL of butanol. These were diluted into seven working solutions for gas chromatography-mass spectrometry (GC-MS) analysis. Fecal samples (25 mg) were processed by adding 500 µL of water containing 0.5% phosphoric acid, followed by freeze-grinding (50 Hz, 3 min × 2), sonication (10 min), and centrifugation (4°C, 13,000 × $g$, 15 min). The supernatant (200 µL) was extracted with 200 µL of butanol containing 10 µg/mL 2-ethylbutyric acid, vortexed (10 s), sonicated (10 min), and centrifuged again (4°C, 13,000 × $g$, 5 min). The final supernatant was transferred to vials for GC-MS analysis. GC-MS analysis was performed using an Agilent 8890B-5977B/7000D instrument with an HP FFAP column and helium as the carrier gas. The temperature program started at 80°C, ramped to 120°C at 20°C/min, then to 160°C at 5°C/min, and held at 220°C for 3 min. Mass spectrometry conditions included an EI ion source with temperatures set at 230°C (ion source) and 150°C (quadrupole), and data were acquired in SIM mode. Quality control (QC) samples were inserted every 5–10 samples to ensure system stability and repeatability, with relative standard deviation (RSD) values for target compounds required to be below 15%.

## Data analysis

The data matrix was uploaded to the Majorbio Cloud Platform for analysis. It was preprocessed using the 80% rule to remove variables with missing values, imputing missing values with the minimum value, normalizing with total sum normalization, and removing variables with RSD >30% in QC samples. The data were then log10-transformed to obtain the final matrix for analysis. Principal component analysis (PCA) and orthogonal partial least squares discriminant analysis (OPLS-DA) were performed using the ropls package in R, with sevenfold cross-validation to assess model stability. Differential metabolites were identified based on VIP >1 and $P < 0.05$ from the OPLS-DA model and Student's $t$-test. Pathway analysis was conducted using the Kyoto Encyclopedia of Genes and Genomes (KEGG) database, and pathway enrichment was performed with scipy stats in Python, identifying relevant biological pathways through Fisher's exact test.

Bioinformatics analysis was performed using the Usearch software (version 7.0) through the Majorbio Cloud Platform to cluster OTUs and identify representative

sequences at a 97% similarity level (19). Fecal microbial alpha and beta diversity were analyzed using Mothur (version v.1.30.1) and Qiime 2, respectively. Microbial structures at the phylum and genus levels were analyzed using R and Python packages (Wilcoxon rank sum test). LEfSe was used to compare differentially abundant taxa with a significance level of 0.05 for the Kruskal-Wallis test and a linear discriminant analysis (LDA) threshold of 3.

Before conducting statistical analysis, the normality and homogeneity of variance of the data were checked using the test procedures of JMP software (JMP, version 10; SAS Institute Inc., Cary, NC). If $P < 0.05$, it is considered significant, and if $0.05 \leq P \leq 0.10$, it may indicate a trend.

## RESULTS

No yak died or was eliminated during the experimental period.

### Growth performance

The average daily feed intake of penned yaks is shown in Table 2. Grazing significantly increased the final body weight and average daily gain of the yak compared to the pen-fed group, with significant differences observed ($P < 0.05$).

### Serum biochemical indicators

As shown in Table 3, compared to the housing group, grazing significantly reduced the levels of serum globulin and urea nitrogen while increasing the level of AST ($P < 0.05$).

### Serum antioxidant indicators

Grazing significantly increased the levels of T-SOD and T-AOC in the serum of yak (Table 4).

### Serum immune indicators

Grazing also enhanced the serum immune levels in yak, including IgA, IgM, and IgG, as well as several inflammatory indicators, such as IL-2, IL-6, TNF-α, and IFN-γ. However, it decreased the serum levels of IL-4 and IL-10 (Table 5).

### Serum hormone indicators

Grazing elevated the levels of the hormone axis indicators in the serum of yak, including GH, IGF-1, and GHRH, while reducing the level of GHIH (Table 6).

### Plasma metabolomics

PCA and partial least squares discriminant analysis (PLS-DA) were used to analyze the plasma metabolite data of yak. The results showed significant differences in plasma metabolites between grazing and penned yak, indicating that the subsequent research was rational and reliable (Fig. 1A and B). The volcano plot of differential metabolites

**TABLE 2** Effect of feeding patterns on the growth performance of yak

| Items | Feeding mode[a] (mean ± SD) | | P-value |
|---|---|---|---|
| | Grazing yak | Housing yak | |
| Initial body weight, kg | 55.79 ± 4.52 | 56.16 ± 4.35 | 0.6053 |
| Final body weight, kg | 75.95 ± 4.70 | 72.21 ± 4.81 | 0.0389 |
| Average daily gain, g/day | 223.98 ± 27.55 | 178.36 ± 26.99 | <0.0001 |
| Average daily feed intake, kg/day | —[b] | 5.51 ± 0.069 | — |

[a]Grazing yak: grazing yak group ($n = 18$); housing yak: housing yak group ($n = 18$). The same applies to the following tables.
[b]"—", indicates that the data is not applicable because the yaks in the grazing group did not undergo feed intake measurements.

**TABLE 3** Effect of feeding patterns on serum biochemical indicators of yak

| Items | Feeding mode (mean ± SD) | | P-value |
| --- | --- | --- | --- |
| | Grazing yak | Housing yak | |
| Total protein, g/L | 70.77 ± 7.14 | 70.95 ± 6.79 | 0.9370 |
| Albumin, g/L | 43.34 ± 5.90 | 40.55 ± 4.44 | 0.1084 |
| Globulin, g/L | 27.44 ± 4.64 | 30.40 ± 4.17 | 0.0450 |
| Blood urea nitrogen, mmol/L | 3.27 ± 0.63 | 6.12 ± 1.35 | <0.0001 |
| Total cholesterol, mmol/L | 3.09 ± 0.53 | 3.51 ± 1.10 | 0.3349 |
| Triglyceride, mmol/L | 1.57 ± 0.75 | 1.37 ± 0.20 | 0.4214 |
| Glucose, mmol/L | 5.25 ± 0.73 | 4.80 ± 0.87 | 0.3179 |
| Alanine aminotransferase, U/L | 98.36 ± 26.11 | 102.41 ± 16.08 | 0.5682 |
| Aspartate aminotransferase, U/L | 45.60 ± 5.30 | 36.43 ± 5.90 | 0.0398 |
| Alkaline phosphatase, U/L | 156.05 ± 17.17 | 163.25 ± 23.70 | 0.6934 |

revealed a total of 1,641 metabolites, of which 301 were significantly increased and 120 were significantly decreased in the grazing group ($P < 0.05$, VIP > 1, FC = 1; Fig. 1C; Table S1). KEGG pathway enrichment analysis on the identified metabolites revealed 58 pathways (Table S2), with the top one pathway being Bile secretion, which involved eight differential metabolites (Fig. 1D; Table S3), including serotonin, topotecan, 15-hydroxynorandrostene-3,17-dione glucuronide, rifampicin, phenethylamine glucuronide, cortisol, cholestane-3,7,12,25-tetrol-3-glucuronide, and salicylic acid.

## Fecal microbiota and short-chain fatty acids

A total of 38 samples were analyzed for their microbial diversity, yielding 2,875,703 optimized sequences, with 1,183,024,729 bases and an average sequence length of 411 bp. In this study, we used the Silva 138/16S Bacteria database and employed the USEARCH 11-uparse algorithm to cluster OTUs at a sequence similarity threshold of 0.97. The classification confidence was set at 0.7, and a total of 5,478 OTUs were obtained, ensuring the quality of the sequencing data and the reliability of the analysis results. As shown in Fig. 1, feeding mode had no significant effect on the α-diversity of the fecal microbiota in yak (Fig. 2A through C). However, grazing significantly altered the β-diversity compared to the penned group (R = 0.6003, $P = 0.0010$, Fig. 2D). At the phylum level, among the top five fecal microbial phyla in yak, grazing significantly increased the relative abundance of Actinobacteriota and decreased the level of Bacteroidota (Fig. 2E; Fig. S1). At the genus level, among the top 20 fecal microbial genera, grazing significantly increased the relative abundance of *Christensenellaceae_R-7_group*, *Monoglobus*, *norank_o_Clostridia_UCG-014*, *Prevotellaceae_UCG-004*, *Romboutsia*, *norank_f_Ruminococcaceae*, and *NK4A214_group*, while decreasing the relative abundance of *Alistipes*, *Prevotellaceae_UCG-003*, and *Phascolarctobacterium* (Fig. 2F; Fig. S2). Additionally, the LEfSe hierarchical species tree (threshold >3) and LDA discriminant results table (threshold >3) showed that there were 74 differential microbial taxa in the fecal microbiota of yak under different feeding modes, from the phylum to genus level (Fig. 3A and B). Among these, grazing yak had 38 dominant microbial taxa in their feces, while penned yak had 36 dominant microbial taxa. Furthermore, BugBase phenotypic prediction showed that grazing increased the Gram-positive, Forms_Biofilms,

**TABLE 4** Effects of different feeding patterns on serum antioxidant indicators in yak

| Items | Feeding mode (mean ± SD) | | P-value |
| --- | --- | --- | --- |
| | Grazing yak | Housing yak | |
| Total superoxide dismutase, U/mL | 470.26 ± 32.00 | 385.29 ± 23.37 | <0.0001 |
| Total antioxidant capacity, U/mL | 49.46 ± 2.62 | 46.61 ± 2.78 | 0.0024 |
| Malondialdehyde, nmol/mL | 209.99 ± 29.65 | 184.43 ± 36.55 | 0.6954 |
| Glutathione peroxidase, U/mL | 9.02 ± 1.63 | 8.66 ± 1.21 | 0.3190 |

**TABLE 5** Effects of different feeding patterns on serum immune indicators in yak

| Items | Feeding mode (mean ± SD) | | P-value |
|---|---|---|---|
| | Grazing yak | Housing yak | |
| Immunoglobulin A, g/L | 1.64 ± 0.05 | 1.32 ± 0.07 | <0.0001 |
| Immunoglobulin M, g/L | 1.37 ± 0.05 | 1.13 ± 0.05 | <0.0001 |
| Immunoglobulin G, g/L | 7.52 ± 0.34 | 6.05 ± 0.30 | <0.0001 |
| Interleukin 2, pg/mL | 426.18 ± 32.42 | 314.91 ± 32.25 | <0.0001 |
| Interleukin 4, pg/mL | 29.85 ± 5.24 | 49.89 ± 5.45 | <0.0001 |
| Interleukin 6, pg/mL | 281.97 ± 22.74 | 199.05 ± 24.86 | <0.0001 |
| Interleukin 10, pg/mL | 93.94 ± 21.26 | 161.83 ± 20.32 | <0.0001 |
| Tumor necrosis factor α, pg/mL | 261.41 ± 21.69 | 193.51 ± 20.56 | <0.0001 |
| Interferon γ, pg/mL | 2,147.50 ± 182.88 | 1,564.69 ± 146.79 | <0.0001 |

and aerobic characteristics of fecal microbiota, while decreasing the anaerobic characteristic (Fig. S3).

Feeding mode significantly altered the content of fecal SCFAs in yaks. Specifically, the levels of SCFAs in the feces of grazing yak were significantly lower than those in penned yak, including acetate, propionate, isobutyrate, butyrate, isovalerate, valerate, isohexanoate, and hexanoate ($P < 0.05$, Fig. 4A). These metabolites were enriched in a total of 20 metabolic pathways through KEGG pathway analysis, with changes occurring in protein digestion and absorption and carbohydrate digestion and absorption, involving acetate, propionate, isobutyrate, butyrate, and isovalerate (Fig. 4B). Additionally, the Mantel test heatmap showed that grazing had a significant correlation with isohexanoate in yak feces (Mantel's $P < 0.05$, Fig. 4C), while it was negatively correlated with acetate, propionate, isobutyrate, butyrate, isovalerate, and valerate ($P < 0.05$). Housing was significantly correlated with propionate, isobutyrate, isovalerate, and valerate (Mantel's $P < 0.05$), and negatively correlated with butyrate and hexanoate.

## DISCUSSION

In this study, grazing yak exhibited significant advantages in final body weight and daily weight gain, which are closely related to the nutritional composition of natural forage and their movement behavior. On the one hand, seasonal changes have a significant impact on grassland ecosystems in the Qinghai-Tibet Plateau. During the warm season, grasses reach their peak growth, with high biodiversity and significantly increased grassland productivity (20, 21). Previous studies have shown that the nutritional components of forage exhibit significant seasonal differences, with higher levels of crude protein, crude ash, and short-chain fatty acids in warm-season forage compared to cold-season forage (22, 23). This difference allows yaks to intake richer nutrients during the warm season, which positively impacts their health and production performance. Moreover, the nutritional components of warm-season forage are more easily digested and absorbed by yak (22, 24, 25). Exercise regulates the secretion of GH, GHRH, GHIH, and IGF-1. Regular aerobic exercise boosts GH levels and IGF-1 synthesis, with IGF-1 mediating most of GH's growth-promoting effect (26, 27). Additionally, movement can stimulate the secretion of GHRH, thereby further promoting the release of GH (28, 29). Moreover, GH stimulates IGF-1 production, which then triggers GHIH release, inhibiting

**TABLE 6** Effects of different feeding patterns on serum growth hormone axis indicators in yak

| Items | Feeding mode (mean ± SD) | | P-value |
|---|---|---|---|
| | Grazing yak | Housing yak | |
| Growth hormone, ng/mL | 15.93 ± 1.09 | 10.51 ± 1.27 | <0.0001 |
| Insulin-like growth factor 1, ng/mL | 389.21 ± 26.89 | 249.67 ± 29.93 | <0.0001 |
| Growth hormone-releasing hormone, pg/mL | 37.38 ± 1.99 | 26.34 ± 1.77 | <0.0001 |
| Growth hormone-inhibiting hormone, pg/mL | 203.90 ± 18.05 | 306.30 ± 16.12 | <0.0001 |

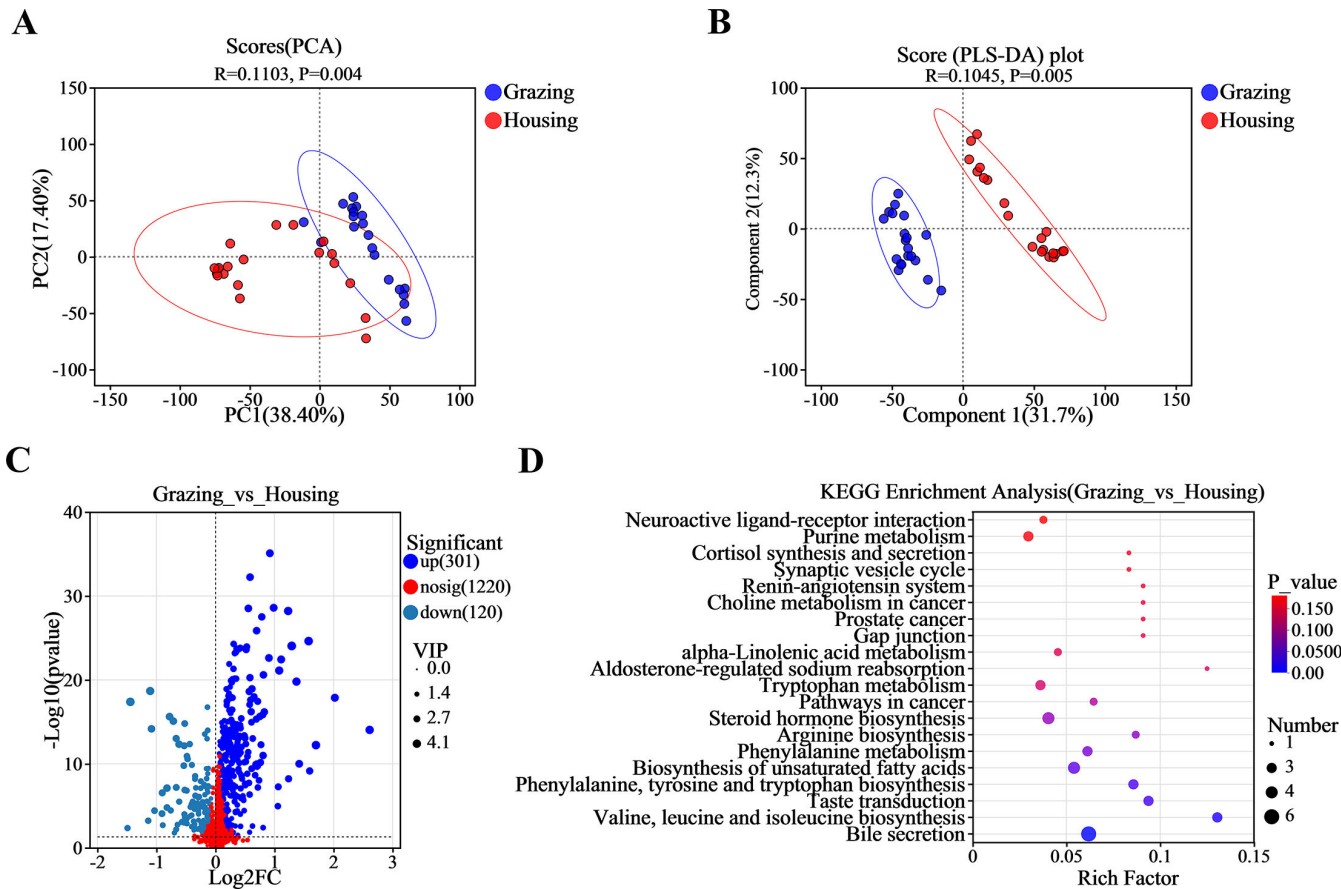

**FIG 1** Plasma metabolome analysis diagrams. (A) PCA score plot; (B) PLS-DA score plot; (C) volcano plot of differential metabolites; and (D) KEGG enrichment plot.

GH and creating a dynamic balance (30, 31). This balance is crucial for the normal secretion of GH (32), which can promote yak growth.

Furthermore, grazing significantly reduced the levels of serum globulin and urea nitrogen in yak while increasing the levels of AST. Serum globulin is an important plasma protein synthesized by the liver, with multiple physiological functions including maintaining colloid osmotic pressure, transporting small molecules, and participating in immune reactions (33, 34). The reduction in serum globulin may be related to the more balanced nutritional intake of natural forage by grazing yak, which reduces metabolic burdens. Urea nitrogen in the blood is generally considered an indirect marker of systemic protein metabolism (35, 36). The decrease in urea nitrogen levels indicates more efficient protein metabolism in the body (37, 38), which may be associated with the higher protein content and better utilization of high-quality forage consumed by yak. The increase in AST levels may reflect the active state of liver metabolism, which helps grazing yaks better regulate the distribution of nutrients within the body (39). This study was conducted from June to October, coinciding with the warm season on the Qinghai-Tibet Plateau, when forage is lush and yaks can graze freely. Compared with housing yak, grazing yak have more freedom of movement, which helps regulate hormone secretion. These factors collectively promote the growth of the yak.

The significant increase in T-SOD and T-AOC in the serum of grazing yak suggests that the intake of active substances, such as polyphenols in natural forages on the Qinghai-Tibet Plateau, has enhanced the oxidative stress defense system (40, 41). Additionally, the increased physical activity of grazing yaks in the natural environment may have promoted the synthesis and activity of antioxidant enzymes in their bodies, thereby better

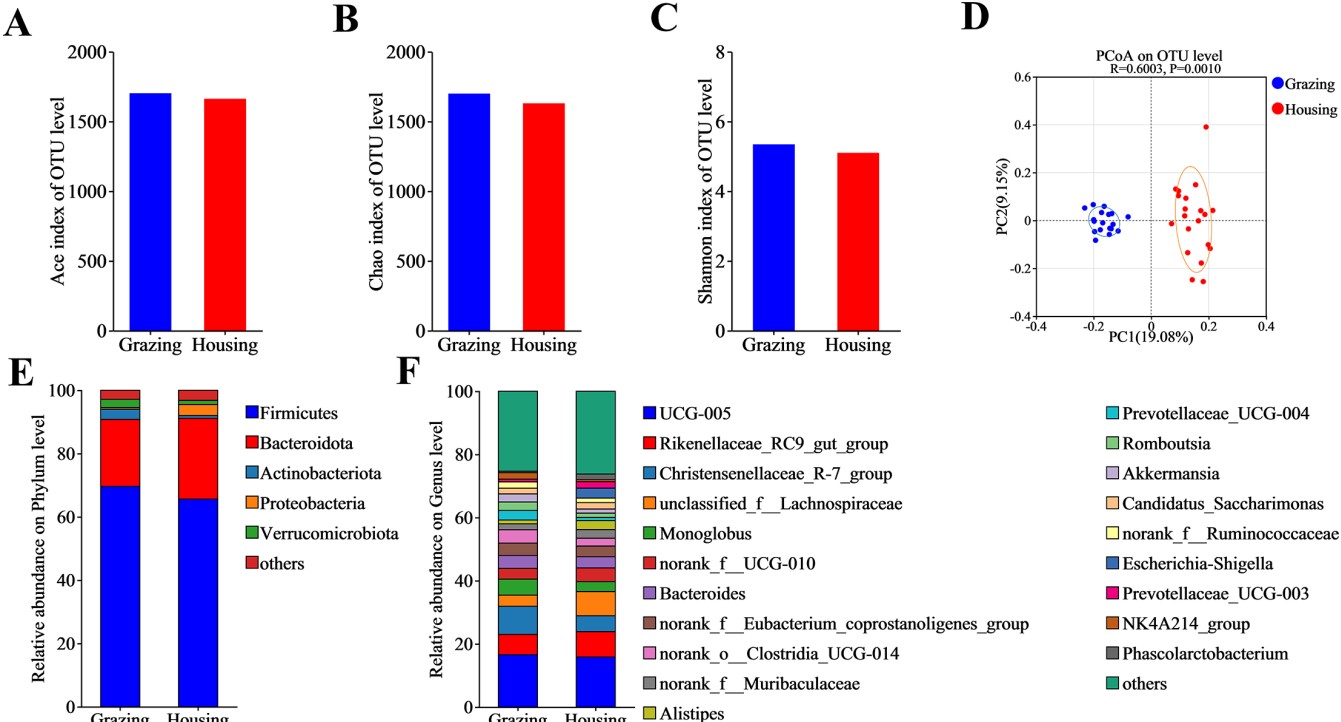

**FIG 2** Alpha diversity, beta diversity, and microbiota composition at phylum and genus levels of fecal microbiota in different treatment groups. (A) ACE index; (B) *Chao1* index; (C) Shannon index; (D) beta diversity; (E) microbiota at phylum level; and (F) microbiota at genus level.

coping with oxidative stress (42, 43). Grazing increased the levels of IgA, IgM, IgG, and some inflammatory indicators (such as IL-2, IL-6, TNF-α, and IFN-γ), while decreasing the levels of IL-4 and IL-10. This indicates that the grazing environment may have enhanced the yak's immune system by providing diverse antigenic stimuli (such as microbial and plant antigens in natural forages). Notably, the upregulation of pro-inflammatory cytokines such as IL-6, TNF-α, and IFN-γ is part of exercise-induced immune activation rather than pathological inflammation (44–46). This activation can reduce the over-release of pro-inflammatory cytokines, thereby achieving immune regulation (47, 48). Plasma metabolomics analysis revealed significant differences in metabolites between grazing and pen-fed yak, especially in bile acid metabolism-related pathways. Certain metabolites in the plasma of grazing yak were significantly upregulated, which may be related to phytochemicals in natural forages that can modulate bile acid metabolism, thereby affecting fat absorption and energy metabolism (49, 50). These changes in metabolites also reflect the interactions between the gut microbiota and host metabolism in grazing yaks. The alterations in bile acid-related metabolites in the plasma metabolome (such as decreased cortisol and increased phenylethylamine glucuronide) are associated with the activation of the farnesoid X receptor signaling pathway regulated by gut microbiota (51, 52). This activation may regulate glucose metabolism by promoting the secretion of glucagon-like peptide-1, which could explain the higher blood glucose levels observed in the grazing group.

In this study, grazing significantly increased the relative abundance of microbial genera in yak feces, including *Christensenellaceae_R-7_group*, *Monoglobus*, *Romboutsia*, *NK4A214_group*, *Prevotellaceae_UCG-004*, *norank_o_Clostridia_UCG-014*, and *norank_f_Ruminococcaceae*. These microbial genera have been confirmed to influence host health by modulating the host immune system and are also known to enhance the host's ability to degrade cellulose (23, 53). For example, *Christensenellaceae_R-7_group* has been shown to promote the secretion of intestinal IgA, thereby enhancing intestinal mucosal immunity (54). These changes in microbial composition may help yaks better

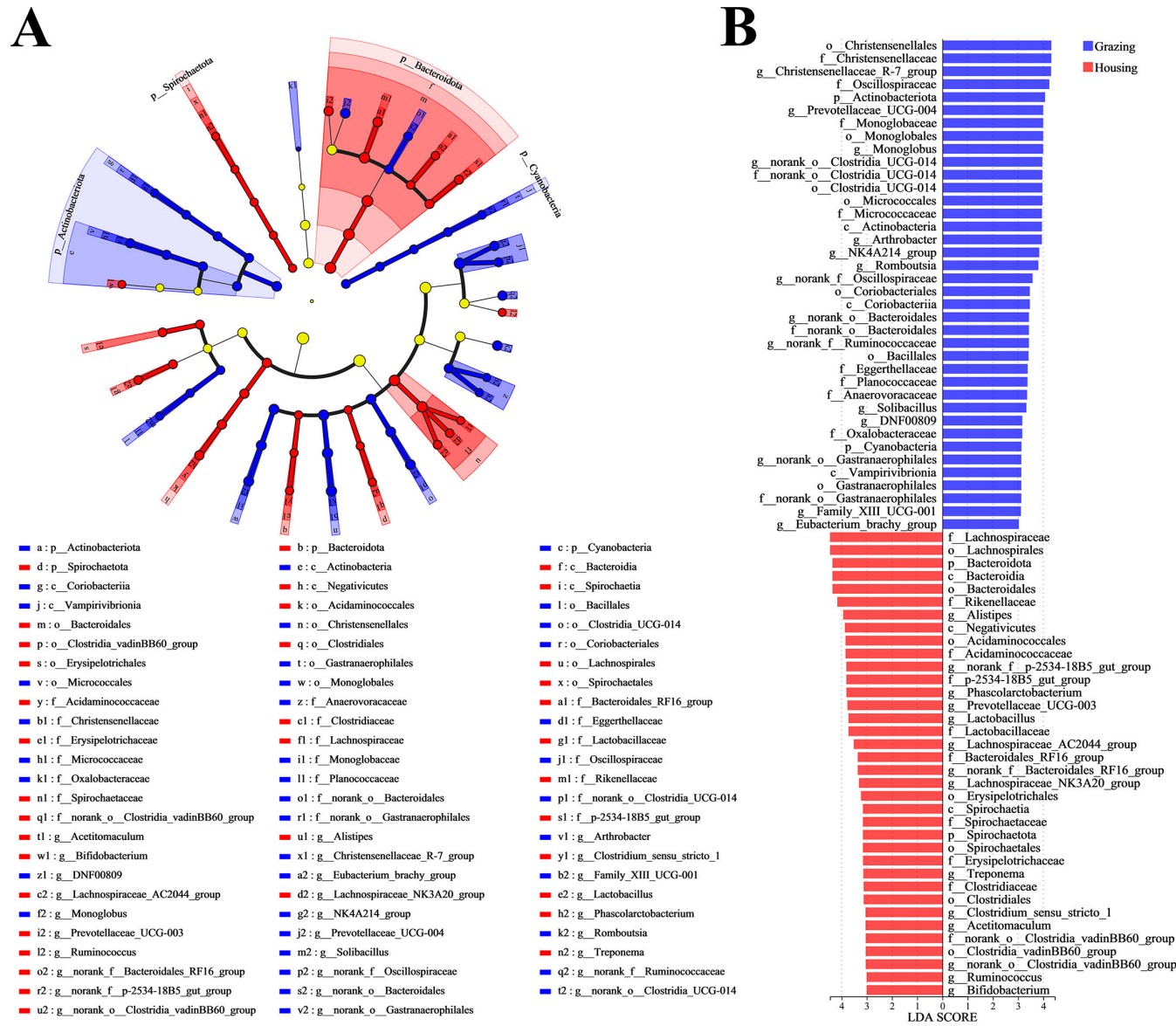

**FIG 3** LEfSe multi-level species differential discriminant analysis. (A) LEfSe multi-level species hierarchical tree diagram and (B) LDA discrimination result table.

adapt to the dietary changes and potential pathogen challenges in the grazing environment, thus maintaining gut health and overall well-being. In contrast, under housing-feeding conditions, with stable feed intake, *Alistipes*, *Prevotellaceae_UCG_003*, and *Phascolarctobacterium* became the dominant microbial genera in the yak gut. This is mainly because the stable supply of feed components provides these genera with abundant substrates, thereby promoting their growth and metabolism to meet the nutritional needs of the yak. Therefore, although the fecal microbial community of grazing yak exhibited higher functional diversity, the total amount of SCFAs in their feces was still lower than that of the housing yak.

The BugBase phenotypic prediction revealed that grazing significantly altered the composition of the fecal microbiota in yak. Specifically, there was an increase in the proportion of Gram-positive bacteria (Gram_Positive), biofilm-forming capacity (Forms_Biofilms), and aerobic bacteria (Aerobic), while the proportion of anaerobic bacteria (Anaerobic) decreased. These changes likely indicate that grazing, by altering the diet and environmental exposure of yak, has reshaped the fecal microbial community (55, 56). The natural forage consumed by grazing yak is rich in phytochemicals,

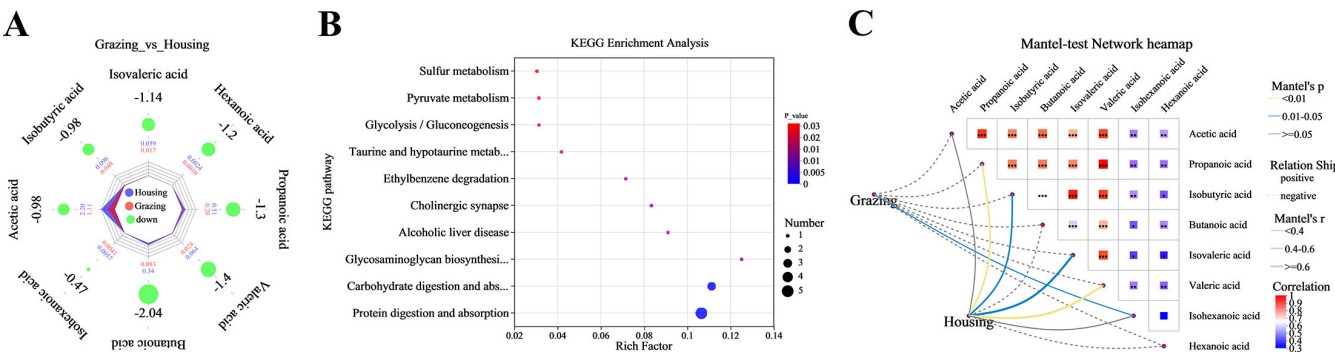

**FIG 4** Radar charts, KEGG enrichment diagrams, and Mantel-test network heatmaps of fecal short-chain fatty acids of feces in different treatment groups. (A) Radar chart; (B) KEGG enrichment diagrams; and (C) Mantel-test network heatmaps.

which can modulate bile acid metabolism and subsequently affect fat absorption and energy metabolism (57). The enhanced ability to form biofilms allows microbes to better immobilize and utilize nutrients in complex environments, while also improving their stress resistance (58, 59). Moreover, the changes in the microbial community due to grazing may accelerate the degradation rate of yak feces, thereby influencing carbon and nutrient cycling (60, 61). Additionally, microbial phenotype prediction showed an increased proportion of aerobic bacteria in the grazing group, which is consistent with the co-evolutionary characteristics of the host-microbiota under hypoxic conditions on the plateau (50, 62). The increase in aerobic bacteria may also inhibit the growth of certain anaerobic pathogens, thereby exerting a positive influence on the intestinal health of the yak. Overall, these adjustments in the microbial community not only help yaks better adapt to high-altitude environments but may also enhance their overall health by modulating the host's metabolism and immune functions.

## Conclusion

By comparing the growth performance, physiological indicators, antioxidant capacity, immune function, hormone levels, metabolome, and microbiome of grazing and pen-fed yak, we revealed the significant benefits of the grazing model for yak in the warm season. Grazing improved the growth performance and nutritional metabolism efficiency of yak, enhanced their antioxidant capacity and immune function, and optimized the structure of the gut microbiota. These results indicate that the grazing model can better utilize natural forage resources to promote yak health and improve production performance. Changes in key metabolic pathways and microbiota suggest potential mechanisms underlying the benefits of grazing on yak health. Future research should further explore these mechanisms, clarify how bioactive compounds in natural forages regulate metabolism and immune function, and investigate the long-term impacts of grazing on yak health and ecosystems, providing a more comprehensive scientific basis for sustainable yak husbandry. This study provides a scientific basis for optimizing yak husbandry models in high-altitude areas and emphasizes the important role of grazing in enhancing yak welfare and ecosystem productivity.

## ACKNOWLEDGMENTS

This work was supported by the National Key Research and Development Program of China (2022YFD1302101) and the Modern Agricultural Industry Technology System for Beef and Yak (CARS-37).

Y.X.: Writing—review and editing, Writing—original draft, Formal analysis, Data curation. Y.C.: Project administration. Z.C.: Investigation, Methodology. D.G.: Validation, Supervision. Q.D.: Visualization. Y.Z.: Writing—review and editing, Supervision, Funding acquisition, Resources. W.B.: Funding acquisition, Conceptualization.

## AUTHOR AFFILIATIONS

[1], Xizang Academy of Agricultural and Animal Husbandry Sciences Institute of Animal Science, Lhasa, Tibet, China

[2]Precision Livestock and Nutrition Unit, TERRA Teaching and Research Center, Gembloux Agro-Bio Tech, University of Liège, Gembloux, Belgium

## AUTHOR ORCIDs

Yining Xie http://orcid.org/0009-0005-8983-7169
Wangdui Basang http://orcid.org/0000-0002-4411-4646
Yanbin Zhu http://orcid.org/0000-0002-7273-2409

## FUNDING

| Funder | Grant(s) | Author(s) |
| --- | --- | --- |
| National Key Research and Development Program of China | 2022YFD1302101 | Wangdui Basang |
| Modern Agricultural Industry Technology System for Beef and Yak | CARS-37 | Yanbin Zhu |

## AUTHOR CONTRIBUTIONS

Yining Xie, Data curation, Formal analysis, Writing – original draft, Writing – review and editing | Yangji Cidan, Project administration | Zhuoma Cisang, Investigation, Methodology | Deji Gusang, Supervision, Validation | Quzha Danzeng, Visualization | Wangdui Basang, Conceptualization, Funding acquisition | Yanbin Zhu, Funding acquisition, Supervision, Writing – review and editing

## DATA AVAILABILITY

The data that support the findings of this study can be found in the supplementary material of this article. The raw sequencing data of the fecal microbiota obtained in this study have been deposited in the NCBI Sequence Read Archive (SRA) database, with the accession number PRJNA1223345 (https://www.ncbi.nlm.nih.gov/bioproject/PRJNA1223345).

## ADDITIONAL FILES

The following material is available online.

### Supplemental Material

**Supplemental figures (Spectrum01001-25-s0001.docx).** Figures S1 to S3.
**Supplemental tables (Spectrum01001-25-s0002.xlsx).** Tables S1 to S3.

### Open Peer Review

**PEER REVIEW HISTORY (review-history.pdf).** An accounting of the reviewer comments and feedback.

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
