## [Reviewer comments · Microbiology Spectrum]

Microbiology Spectrum

Effects of warm-season feeding on yak growth, antioxidant capacity, immune function, and fecal microbiota

Yining Xie, Yangji Cidan, Zhuoma Cisang, Deji Gusang, Quzha Danzeng, Wangdui Basang, and yanbin Zhu

Corresponding Author(s): yanbin Zhu, Gansu Agricultural University

Review Timeline:

Submission Date:	April 3, 2025
Editorial Decision:	May 6, 2025
Revision Received:	May 10, 2025
Editorial Decision:	May 18, 2025
Revision Received:	May 21, 2025
Accepted:	May 22, 2025

Editor: Jinshui Lin

Reviewer(s): Disclosure of reviewer identity is with reference to reviewer comments included in decision letter(s). The following individuals involved in review of your submission have agreed to reveal their identity: Lizhuang Hao (Reviewer #1); Tariq Shah (Reviewer #2)

Transaction Report:

DOI: <https://doi.org/10.1128/spectrum.01001-25>

Re: Spectrum01001-25 (**Effects of Warm-Season Feeding Patterns on Growth Performance, Antioxidant Capacity, Immune Function, Metabolome, and Fecal Microbiota in Yaks**)

Dear Dr. yanbin Zhu:

Thank you for the privilege of reviewing your work. Below you will find my comments, instructions from the Spectrum editorial office, and the reviewer comments.

Reviewer 2 left me a message suggesting that the authors further clarify the following points. Please respond as well.

1. **Statistical Methods:** A more detailed explanation of the statistical methods used, including how the sample size was determined, would strengthen the rigor of the analysis and ensure the robustness of the results.
2. **Treatment Clarification:** It would be important to confirm whether the yaks received any treatments (e.g., antibiotics, anthelmintics, vaccines) during the trial, as these could potentially affect the gut microbiota results and overall interpretation of the findings.
3. **Sequencing Data Quality:** I recommend that the authors confirm the consistency and quality of the sequencing data, and include a more detailed description of sequencing depth, OTUs operational taxonomic units count, and detection limits in the methods section to enhance transparency.

Revision Guidelines

Sincerely,
Jinshui Lin
Editor
Microbiology Spectrum

Reviewer #1 (Comments for the Author):

The manuscript titled "Effects of Warm-Season Feeding Patterns on Growth Performance, Antioxidant Capacity, Immune Function, Metabolome, and Fecal Microbiota in Yaks" examines the impacts of grazing versus pen-feeding on yaks during the warm season on the Tibetan Plateau. The study finds that grazing significantly enhances growth performance (increased final body weight and average daily gain), antioxidant capacity (higher T-SOD and T-AOC levels), and immune function (elevated immunoglobulins and pro-inflammatory cytokines). It also optimizes the gut microbiota structure and alters plasma metabolite profiles, particularly in bile acid metabolism pathways. These findings suggest that grazing is a more effective feeding strategy for yaks, promoting better health and production performance by utilizing natural forage resources.

The comments for this manuscript are listed.

Minor:

1. The author should carefully check the Latin terms in the manuscript, which should be italicized, such as "*Bos grunniens*" (line 45).

2. Consistency of Terminology: The author needs to standardize the use of specialized terms in the text. For example, expressions such as "penned fed," "Housing yak," and "pen-feeding" should be consistent to avoid confusion.

3. At line 94, "yaks" should be in the singular form.

4. Microorganisms at the phylum level do not need to be italicized, such as "Actinobacteriota" at line 250 and "Bacteroidota" at line 251. The author should also carefully check the rest of the manuscript for consistency.

5. At line 192, " $p < 0.05$ " should be capitalized as " $P < 0.05$ ".

6. The author should refer to the journal's format and revise the citation style of the manuscript.

In summary, this article performs outstandingly in terms of research methods, data analysis, and conclusions. It is recommended to consider and implement the above suggestions to improve the overall quality of the article.

Reviewer #2 (Public repository details (Required)):

The large datasets must be deposited in a public repository such as NCBI, GenBank or another appropriate platform.

Reviewer #2 (Comments for the Author):

The study titled "Effects of Warm-Season Feeding Patterns on Growth Performance, Antioxidant Capacity, Immune Function, Metabolome, and Fecal Microbiota in Yaks" presents a well conducted and insightful analysis. It effectively integrates growth performance, immune function, metabolomics, and gut microbiota, making it highly relevant to yak husbandry and sustainable livestock production under climate change. However I suggest the following improvements to enhance clarity, scientific rigor, and readability follows as:

1. Title and Abstract: The title is informative but slightly lengthy. Consider simplifying it by grouping related terms or focusing on the key outcomes. The abstract is comprehensive but dense. Breaking up long sentences and reducing redundancy would enhance readability. Ensure *Bos grunniens* is italicized. Use the term "yaks" instead of "yak" for consistency and grammatical correctness.

2. Introduction: The introduction provides a well-rounded background on yak biology, ecological importance, and evolving husbandry practices. The rationale for comparing grazing and pen-feeding systems is presented. The hypothesis is logical but could be rephrased in a more concise and testable format. For example: "We hypothesize that warm-season grazing improves metabolic efficiency and immune responses compared to pen-feeding." Consider including one or two guiding research questions, such as: How do grazing and pen feeding differ in terms of long-term sustainability for yak populations, particularly under the pressure of climate change? Do shifts in gut microbiota in grazing yaks enhance metabolism or disease resistance?

3. Materials and Methods. Please clarify whether the yaks received any antibiotic, anthelmintic, or vaccine treatments before or during the trial. This is critical for interpreting the microbiome and metabolomics data. Justify the selection of the V3-V4 region for 16S rRNA sequencing. Include basic sequencing statistics such as sequencing depth, OTUs count, and coverage to enhance transparency. The extraction and analysis methods are sound; however, including detection limits and rationale for the choice of solvents would increase methodological rigor.

4. Results The results are structured. Including descriptive statistics (e.g., mean {plus minus} SD) alongside p-values or significance indicators will improve interpretability. Try linking specific biomarker changes (e.g immune cytokines or metabolites) to physiological processes such as liver function, nutrient metabolism, or muscle activity.

5. Discussion: The discussion is comprehensive and well linked to the study objectives. Consider being more concise by focusing on the key findings and reducing redundancy, especially in sections about hormone regulation and immune responses. The integration of microbiota and metabolomics is a major strength this section would benefit from streamlining to highlight the

novelty and significance of these findings.

6. Conclusion. The conclusion is generally satisfactory. It would be stronger if it briefly mentioned the underlying mechanisms and offered clear future research directions based on the studys findings.

6. Data availability: The data availability statement should be more specific. Please include an accession number (e.g. NCBI SRA) or provide a link to an online repository if available for better transparency.

**Effects of Warm-Season Feeding Patterns on Growth Performance, Antioxidant**

**Capacity, Immune Function, Metabolome, and Fecal Microbiota in Yaks**

Yining Xie^{ab}, Yangji Cidan^a, Zhuoma Cisang^a, Deji Gusang^a, Quzha Danzeng^a,

Wangdui Basang^{a#}, Yanbin Zhu^{a#}

5 ^a *Xizang Academy of Agriculture and Animal Husbandry Sciences, Lhasa 850000,*

*China*

7 ^b *Precision Livestock and Nutrition Unit, TERRA Teaching and Research Center,*

*Gembloux Agro-Bio Tech, University of Liège, Gembloux, Belgium*

[#]Corresponding author: Yanbin Zhu Xizang Academy of Agriculture and Animal

Husbandry Sciences, Lhasa 850000, *China*; Email: zhuyanbin126@126.com.

Wangdui Basang Xizang Academy of Agriculture and Animal Husbandry Sciences,

Lhasa 850000, *China*; Email: bw0891@163.com

Co-first author: Yining Xie, yining.xie@student.uliege.be; Yangji Cidan,

13889092363@163.com. The first 2 authors contributed equally to this work.

**ABSTRACT**

**BACKGROUND:** The yak (*Bos grunniens*) is of great importance to the local
ecosystem and animal husbandry on the Tibetan Plateau. However, the impacts of
changes in feeding patterns due to modern animal husbandry on yak growth, health,
and ecosystem interactions remain unclear. This study investigates the effects of
warm-season grazing and pen-feeding on yak growth performance, antioxidant
capacity, immune function, metabolome, and fecal microbiota.

**RESULTS:** The study found that grazing significantly increased the final body
weight and average daily gain of yaks ($P < 0.05$), reduced serum globulin and urea
nitrogen levels, and elevated aspartate aminotransferase (AST) levels. Grazing also
significantly enhanced serum total superoxide dismutase (T-SOD) and total
antioxidant capacity (T-AOC), increased levels of immunoglobulins (IgA, IgM, IgG)
and pro-inflammatory cytokines (IL-2, IL-6, TNF- α , IFN- γ), while decreasing IL-4
and IL-10 levels. Additionally, grazing significantly altered the plasma metabolite
profile, particularly in bile acid metabolism pathways. The relative abundance of
beneficial microbial genera (e.g., *Christensenellaceae_R-7_group*, *Monoglobus*,
*Romboutsia*) in the feces of grazing yaks was significantly higher, while total
short-chain fatty acids (SCFAs) were lower than in penned yaks.

**CONCLUSION:** Grazing significantly improved growth performance and nutritional
metabolism efficiency, enhanced antioxidant and immune functions, and optimized
the structure of the gut microbiota in yaks. These findings indicate that grazing can

better utilize natural forage resources to promote yak health and improve production

performance.

**Keywords:** Yak, feeding pattern, immune function, fecal microbiota

INTRODUCTION

The yak (*Bos grunniens*), a unique ruminant native to the Tibetan Plateau, serves
as the cornerstone of local animal husbandry by providing meat, milk, fiber, and
essential ecosystem services (Wang et al., 2013; Xin et al., 2019). As one of the few
domesticated ungulates capable of thriving at altitudes above 3000 meters, yaks have
evolved remarkable physiological and metabolic adaptations to withstand extreme
environmental pressures, including hypoxia, diurnal temperature fluctuations, and
seasonal forage shortages (Qiu et al., 2012; Ayalew et al., 2021; Ahmad et al., 2024).
However, yak husbandry systems are undergoing significant changes due to
increasing human activities and the impacts of climate change on alpine meadows
(Krishnan et al., 2016; Mipam et al., 2019; Sapkota et al., 2022). The rapid
development of modern animal husbandry has introduced new feeding practices, such
as pen-feeding and supplementary feeding (Vaintrub et al., 2021; Yi et al., 2023; He
et al., 2024). These practices are designed to optimize yak growth rates, improve feed
utilization, and reduce environmental disturbances (Xue et al., 2021; Zou et al., 2021;
Dai et al., 2022). Despite these advancements, grazing remains an extremely
important production method in yak husbandry on the Tibetan Plateau. This system
takes advantage of the seasonal abundance of grassland resources, especially during
the warm season (May to September), when monsoon rains promote vigorous plant
growth and increase the nutritional value of forage (Ma et al., 2019; Yang et al.,
2024b). In traditional grazing, yaks roam freely over vast grasslands, exploiting
natural forage resources, which is crucial for various aspects of their biology,

including growth, health, and metabolism (Jing et al., 2022; Sapkota et al., 2022; Shah
et al., 2023).

[revised manuscript text omitted]

**Serum hormone indicators**

Grazing elevated the levels of the hormone axis indicators in the
serum of yaks, including GH, IGF-1, and GHRH, while reducing the level
of GHIH (Table 6).

**Plasma Metabolomics**

Principal component analysis (PCA) and partial least squares discriminant
analysis (PLS-DA) were used to analyze the plasma metabolite data of yaks. The
results showed significant differences in plasma metabolites between grazing and
penned yaks, indicating that the subsequent research was rational and reliable (Fig.
1A-B). The volcano plot of differential metabolites revealed a total of 1641
metabolites, of which 301 were significantly increased and 120 were significantly
decreased in the grazing group ($P < 0.05$, $VIP > 1$, $FC = 1$; Fig. 1C, Supplementary
materials, Table 1). KEGG pathway enrichment analysis on the identified metabolites
revealed 58 pathways (Supplementary materials, Table 2), with the top 1 pathway

being Bile secretion, which involved 8 differential metabolites (Fig. 1D,
Supplementary materials, Table 3), including Serotonin, Topotecan,
15-Hydroxynorandrostene-3,17-Dione Glucuronide, Rifampicin, Phenethylamine
Glucuronide, Cortisol, Cholestane-3,7,12,25-Tetrol-3-Glucuronide, and Salicylic
Acid.

**Fecal microbiota and short-chain fatty acids**

A total of 38 samples were analyzed for their microbial diversity, yielding
2,875,703 optimized sequences, with 1,183,024,729 bases and an average sequence
length of 411 bp. As shown in Figure 1, feeding mode had no significant effect on the
α -diversity of the fecal microbiota in yaks (Fig. 2A-C). However, grazing
significantly altered the β -diversity compared to the penned group ($R=0.6003$,
$P=0.0010$, Fig. 2D). At the phylum level, among the top 5 fecal microbial phyla in
yaks, grazing significantly increased the relative abundance of *Actinobacteriota* and
decreased the level of *Bacteroidota* (Fig. 2E, Supplementary material, Figure 1). At
the genus level, among the top 20 fecal microbial genera, grazing significantly
increased the relative abundance of *Christensenellaceae_R-7_group*, *Monoglobus*,
*norank_o_Clostridia_UCG-014*, *Prevotellaceae_UCG-004*, *Romboutsia*,
*norank_f_Ruminococcaceae*, and *NK4A214_group*, while decreasing the relative
abundance of *Alistipes*, *Prevotellaceae_UCG-003*, and *Phascolarctobacterium* (Fig.
2F, Supplementary material, Figure 2). Additionally, the LEfSe hierarchical species
tree (threshold >3) and LDA discriminant results table (threshold >3) showed that
there were 74 differential microbial taxa in the fecal microbiota of yaks under

different feeding modes, from the phylum to genus level (Fig. 3A-B). Among these,
grazing yaks had 38 dominant microbial taxa in their feces, while penned yaks had 36
dominant microbial taxa. Furthermore, BugBase phenotypic prediction showed that
grazing increased the Gram-Positive, Forms_Biofilms, and Aerobic characteristics of
fecal microbiota, while decreasing the Anaerobic characteristic (Supplementary
material, Figure 3).

Feeding mode significantly altered the content of fecal SCFAs in yaks.
Specifically, the levels of SCFAs in the feces of grazing yaks were significantly lower
than those in penned yaks, including acetate, propionate, isobutyrate, butyrate,
isovalerate, valerate, isohexanoate, and hexanoate ($P < 0.05$, Fig. 4A). These
metabolites were enriched in a total of 20 metabolic pathways through KEGG
pathway analysis, with changes occurring in Protein digestion and absorption and
Carbohydrate digestion and absorption, involving acetate, propionate, isobutyrate,
butyrate, and isovalerate (Fig. 4B). Additionally, The Mantel Test heatmap showed
that grazing had a significant correlation with isohexanoate in yak feces (Mantel's
$P < 0.05$, Fig. 4C), while it was negatively correlated with acetate, propionate,
isobutyrate, butyrate, isovalerate, and valerate ($P < 0.05$). Housing was significantly
correlated with propionate, isobutyrate, isovalerate, and valerate (Mantel's $P < 0.05$),
and negatively correlated with butyrate and hexanoate.

**DISCUSSION**

In this study, grazing yaks exhibited significant advantages in final body weight
and daily weight gain, which are closely related to the nutritional composition of

natural forage and their movement behavior. On the one hand, seasonal changes have
a significant impact on grassland ecosystems in the Qinghai-Tibet Plateau. During the
warm season, grasses reach their peak growth, with high biodiversity and significantly
increased grassland productivity (Li et al., 2023b; Huang et al., 2024). Previous
studies have shown that the nutritional components of forage exhibit significant
seasonal differences, with higher levels of crude protein, crude ash, and short-chain
fatty acids in warm-season forage compared to cold-season forage (Long et al., 1999;
Guo et al., 2021). This difference allows yaks to intake richer nutrients during the
warm season, which positively impacts their health and production performance.
Moreover, the nutritional components of warm-season forage are more easily digested
and absorbed by yaks (Long et al., 1999; Li et al., 2023a; Liu et al., 2023). On the
other hand, exercise can regulate the secretion of GH, GHRH, GHIH, and IGF-1.
Research shows that regular aerobic exercise can increase the basal level of growth
hormone and promote the synthesis of IGF-1. IGF-1 is the main downstream effector
molecule of growth hormone and mediates most of its growth-promoting effects
(Frystyk, 2010; Sabag et al., 2021). Additionally, movement can stimulate the
secretion of GHRH, thereby further promoting the release of GH (Stanley and
Grinspoon, 2015; Hymer and Kraemer, 2023). Moreover, GH secretion promotes the
production of IGF-1, which in turn stimulates the secretion of GHIH, inhibiting GH
secretion and forming a dynamic regulatory balance (Okada and Kopchick, 2001;
Nijenhuis-Noort et al., 2024). This balance is crucial for the normal secretion of GH;
when GHRH secretion increases, GHIH secretion decreases, and GH levels rise, and

vice versa (Lu et al., 2019). Furthermore, grazing significantly reduced the levels of
serum globulin and urea nitrogen in yaks while increasing the levels of AST. Serum
globulin is an important plasma protein synthesized by the liver, with multiple
physiological functions including maintaining colloid osmotic pressure, transporting
small molecules, and participating in immune reactions (Belinskaia et al., 2021; Li et
al., 2021). The reduction in serum globulin may be related to the more balanced
nutritional intake of natural forage by grazing yaks, which reduces metabolic burdens.
Urea nitrogen in the blood is generally considered an indirect marker of systemic
protein metabolism (Eggum, 1970; Mao et al., 2021). The decrease in urea nitrogen
levels indicates more efficient protein metabolism in the body (Figuroa et al., 2002;
Yu et al., 2017), which may be associated with the higher protein content and better
utilization of high-quality forage consumed by yaks. The increase in AST levels may
reflect the active state of liver metabolism, which helps grazing yaks better regulate
the distribution of nutrients within the body (Zhang et al., 2024). This study was
conducted from June to October, coinciding with the warm season on the
Qinghai-Tibet Plateau, when forage is lush and yaks can graze freely. Compared with
housing yaks, grazing yaks have more freedom of movement, which helps regulate
hormone secretion. These factors collectively promote the growth of yaks.

The significant increase in T-SOD and T-AOC in the serum of grazing yaks
suggests that the intake of active substances, such as polyphenols in natural forages on
the Qinghai-Tibet Plateau, has enhanced the oxidative stress defense system (Hao et
al., 2018; Deng et al., 2022). Additionally, the increased physical activity of grazing

yaks in the natural environment may have promoted the synthesis and activity of
antioxidant enzymes in their bodies, thereby better coping with oxidative stress (Daud
et al., 2022; Powers et al., 2023). Grazing significantly increased the levels of
immunoglobulins IgA, IgM, IgG, and some inflammatory indicators (such as IL-2,
IL-6, TNF- α , and IFN- γ), while decreasing the levels of IL-4 and IL-10. This
indicates that the grazing environment may have enhanced the yaks' immune system
by providing diverse antigenic stimuli (such as microbial and plant antigens in natural
forages). Notably, the upregulation of pro-inflammatory cytokines such as IL-6,
TNF- α , and IFN- γ is part of exercise-induced immune activation rather than
pathological inflammation (Syu et al., 2012; Coelho et al., 2016; Luo et al., 2024).
This "physiological inflammation" activates the TLR4/NF- κ B pathway to enhance
macrophage function, while exercise inhibits the NF- κ B signaling pathway to reduce
the over-release of pro-inflammatory cytokines, thereby achieving immune
regulation (Capece et al., 2022; Liu et al., 2024). Plasma metabolomics analysis
revealed significant differences in metabolites between grazing and pen-fed yaks,
especially in bile acid metabolism-related pathways. Certain metabolites in the plasma
of grazing yaks were significantly upregulated, which may be related to
phytochemicals in natural forages that can modulate bile acid metabolism, thereby
affecting fat absorption and energy metabolism (Wang et al., 2024b; Zheng et al.,
2024). These changes in metabolites also reflect the interactions between the gut
microbiota and host metabolism in grazing yaks. The alterations in bile acid-related
metabolites in the plasma metabolome (such as decreased cortisol and increased

phenylethylamine glucuronide) are associated with the activation of the farnesoid X
receptor (FXR) signaling pathway regulated by gut microbiota (Hu et al., 2022; Li et
al., 2024). This activation may regulate glucose metabolism by promoting the
secretion of glucagon-like peptide-1 (GLP-1), which could explain the higher blood
glucose levels observed in the grazing group.

In this study, grazing significantly increased the relative abundance of microbial
genera in yak feces, including *Christensenellaceae_R-7_group*, *Monoglobus*,
*Romboutsia*, *NK4A214_group*, *Prevotellaceae_UCG-004*,
*norank_o_Clostridia_UCG-014*, and *norank_f_Ruminococcaceae*. These microbial
genera have been confirmed to influence host health by modulating the host immune
system and are also known to enhance the host's ability to degrade cellulose (Guo et
al., 2021; Fonseca et al., 2023). For example, *Christensenellaceae_R-7_group* has
been shown to promote the secretion of intestinal IgA, thereby enhancing intestinal
mucosal immunity (Wang et al., 2024a). These changes in microbial composition may
help yaks better adapt to the dietary changes and potential pathogen challenges in the
grazing environment, thus maintaining gut health and overall well-being. In contrast,
under pen-feeding conditions, with stable feed intake, *Alistipes*,
*Prevotellaceae_UCG_003*, and *Phascolarctobacterium* became the dominant
microbial genera in the yak gut. This is mainly because the stable supply of feed
components provides these genera with abundant substrates, thereby promoting their
growth and metabolism to meet the nutritional needs of yaks. Therefore, although the
fecal microbial community of grazing yaks exhibited higher functional diversity, the

total amount of short-chain fatty acids (SCFAs) in their feces was still lower than that
of housing yaks.

The BugBase phenotypic prediction revealed that grazing significantly altered
the composition of the fecal microbiota in yaks. Specifically, there was an increase in
the proportion of Gram-positive bacteria (Gram_Positive), biofilm-forming capacity
(Forms_Biofilms), and aerobic bacteria (Aerobic), while the proportion of anaerobic
bacteria (Anaerobic) decreased. These changes likely indicate that grazing, by altering
the diet and environmental exposure of yaks, has reshaped the fecal microbial
community (Huhe et al., 2017; Zhu et al., 2022). The natural forage consumed by
grazing yaks is rich in phytochemicals, which can modulate bile acid metabolism and
subsequently affect fat absorption and energy metabolism (Zhu et al., 2024). The
enhanced ability to form biofilms allows microbes to better immobilize and utilize
nutrients in complex environments, while also improving their stress resistance (Wu
et al., 2022; Yang et al., 2024a). Moreover, the changes in the microbial community
due to grazing may accelerate the degradation rate of yak feces, thereby influencing
carbon and nutrient cycling (Zhao et al., 2017; Zhang et al., 2023). Additionally,
microbial phenotype prediction showed an increased proportion of aerobic bacteria in
the grazing group, which is consistent with the co-evolutionary characteristics of the
host-microbiota under hypoxic conditions on the plateau (Hao et al., 2024; Wang et
al., 2024b). The increase in aerobic bacteria may also inhibit the growth of certain
anaerobic pathogens, thereby exerting a positive influence on the intestinal health of
yaks. Overall, these adjustments in the microbial community not only help yaks better

adapt to high-altitude environments but may also enhance their overall health by
modulating the host's metabolism and immune functions.

**CONCLUSION**

This study, by comparing the growth performance, physiological indicators,
antioxidant capacity, immune function, hormone levels, metabolome, and microbiome
of grazing and pen-fed yaks, revealed the significant benefits of the grazing model for
yaks in warm season. Grazing significantly improved the growth performance and
nutritional metabolism efficiency of yaks, enhanced their antioxidant capacity and
immune function, and optimized the structure of the gut microbiota. These results
indicate that the grazing model can better utilize natural forage resources to promote
yak health and improve production performance. This study provides a scientific basis
for optimizing yak husbandry models in high-altitude areas and emphasizes the
important role of grazing in enhancing yak welfare and ecosystem productivity.

**CONFLICT OF INTEREST**

The authors state that there are no relevant conflicts of interest with respect to
this article.

**CRedit authorship contribution statement**

**Yining Xie:** Writing – review & editing, Writing – original draft, Formal
analysis, Data curation. **Zhuoma Cisang:** Investigation, Methodology . **Deji Gusang:**
Validation, Supervision. **Quzha Danzeng:** Visualization. **Yangji Cidan:** Project
administration. **Yanbin Zhu:** Writing – review & editing, Supervision, Funding
acquisition, Resources. **Wangdui Basang:** Funding acquisition, Conceptualization.

**FUNDING INFORMATION**

This work was supported by the National Key Research and Development
Program of China (2022YFD1302101) and the Modern Agricultural Industry
Technology System for Beef and Yak (CARS-37).

**DATA AVAILABILITY STATEMENT**

The data that support the findings of this study can be found in the
supplementary material of this article.

**SUPPORTING INFORMATION**

The supporting information can be found in the online version of this article.

**REFERENCE**

- Ahmad, H. I., S. Mahmood, M. Hassan, M. Sajid, I. Ahmed, B.
Shokrollahi, A. H. Shahzad, S. Abbas, S. Raza, K. Khan, S. A. Muhammad, D.
Fouad, F. S. Ataya, and Z. Li. 2024. Genomic insights into Yak (*Bos*
*grunniens*) adaptations for nutrient assimilation in high-altitudes. *Scientific*
*Reports* 14(1):5650. doi: 10.1038/s41598024557123
- Ayalew, W., M. Chu, C. Liang, X. Wu, and P. Yan. 2021. Adaptation
Mechanisms of Yak (*Bos grunniens*) to High-Altitude Environmental Stress.
*Animals (Basel)* 11(8)doi: 10.3390/ani11082344
- Belinskaia, D. A., P. A. Voronina, V. I. Shmurak, R. O. Jenkins, and N.
436 V. Goncharov. 2021. Serum Albumin in Health and Disease: Esterase,
Antioxidant, Transporting and Signaling Properties. *International Journal of*
*Molecular Sciences* 22(19)doi: 10.3390/ijms221910318
- Capece, D., D. Verzella, I. Flati, P. Arboretto, J. Cornice, and G.
Franzoso. 2022. NF- κ B: blending metabolism, immunity, and inflammation.
*Trends in Immunology* 43(9):757-775. doi: 10.1016/j.it.2022.07.004
- Coelho, W. S., L. V. d. Castro, E. Deane, A. Magno-França, A. Bassini,
and L. C. Cameron. 2016. Investigating the Cellular and Metabolic Responses
of World-Class Canoeists Training: A Sportomics Approach. *Nutrients*
8(11)doi: 10.3390/nu8110719
- Dai, D., K. Pang, S. Liu, X. Wang, Y. Yang, S. Chai, and S. Wang. 2022.
Effects of Concentrate Supplementation on Growth Performance, Rumen
Fermentation, and Bacterial Community Composition in Grazing Yaks during
the Warm Season. *Animals* 12(11):1398.

Daud, D. M. A., F. Ahmedy, D. M. P. Baharuddin, and Z. A. Zakaria.
2022. Oxidative Stress and Antioxidant Enzymes Activity after Cycling at
Different Intensity and Duration. *Applied Sciences* 12(18):9161.

Deng, K., J. Ouyang, N. Hu, J. Meng, C. Su, J. Wang, and H. Wang.
2022. Improved colorimetric analysis for subtle changes of powdered
anthocyanins extracted from *Lycium ruthenicum* Murr. *Food Chemistry* doi:
10.1016/j.foodchem.2021.131080

Edgar, R. C. 2013. UPARSE: highly accurate OTU sequences from
microbial amplicon reads. *Nature Methods* doi: 10.1038/nmeth.2604

Eggum, B. O. 1970. Blood urea measurement as a technique for assessing
protein quality. *Br J Nutr* 24(4):983-988. doi: 10.1079/bjn19700101

Figueroa, J. L., A. J. Lewis, P. S. Mille, R. L. Fische, R. S. Gómez, and R.
462 M. Diedrichsen. 2002. Nitrogen metabolism and growth performance of gilts
fed standard corn-soybean meal diets or low-crude protein, amino
acid-supplemented diets. *J Anim Sci* 80(11):2911-2919. doi:
10.2527/2002.80112911x

Fonseca, P. A. d. S., S. Lam, Y. Chen, S. M. Waters, L. L. Guan, and Á.
Cánovas. 2023. Multi-breed host rumen epithelium transcriptome and
microbiome associations and their relationship with beef cattle feed efficiency.
*Scientific Reports* doi: 10.1038/s41598023430978

Frystyk, J. 2010. Exercise and the growth hormone-insulin-like growth
factor axis. *Medicine & Science in Sports & Exercise* 42(1):58-66. doi:
10.1249/MSS.0b013e3181b07d2d

Guo, N., Q. Wu, F. Shi, J. Niu, T. Zhang, A. A. Degen, Q. Fang, L. Ding,
Z. Shang, Z. Zhang, and R. Long. 2021. Seasonal dynamics of diet–gut
microbiota interaction in adaptation of yaks to life at high altitude. *npj*
*Biofilms and Microbiomes* doi: 10.1038/s41522021002076

Hao, D., H. Niu, Q. Zhao, J. Shi, C. An, S. Wang, C. Zhou, S. Chen, Y.
Fu, Y. Zhang, and Z. He. 2024. Impact of high-altitude acclimatization and
de-acclimatization on the intestinal microbiota of rats in a natural high-altitude
environment. *Frontiers in Microbiology* 15:1371247. doi:
10.3389/fmicb.2024.1371247

Hao, D. C., P. G. Xiao, and C. Liu. 2018. Traditional Tibetan medicinal
plants: a highlighted resource for novel therapeutic compounds. *Future*
*Medicinal Chemistry* 10(21):2537-2555. doi: 10.4155/fmc20180235

He, S., Z. Yuan, S. Dai, Z. Wang, S. Zhao, R. Wang, Q. Li, H. Mao, and
D. Wu. 2024. Intensive feeding alters the rumen microbiota and its
fermentation parameters in natural grazing yaks. *Frontiers in Veterinary*
*Science* 11:1365300. doi: 10.3389/fvets.2024.1365300

Hu, W., W. Gao, Z. Liu, Z. Fang, H. Wang, J. Zhao, H. Zhang, W. Lu,
and W. Chen. 2022. Specific Strains of *Faecalibacterium prausnitzii*
Ameliorate Nonalcoholic Fatty Liver Disease in Mice in Association with Gut
Microbiota Regulation. *Nutrients* 14(14)doi: 10.3390/nu14142945

Huang, X., G. Luo, Z. Ma, B. Yao, Y. Du, and Y. Yang. 2024. Modeling
the effect of grazing on carbon and water use efficiencies in grasslands on the
Qinghai–Tibet Plateau. *BMC Ecology and Evolution* 24(1):26. doi:
10.1186/s12862024022154

Huhe, X. Chen, F. Hou, Y. Wu, and Y. Cheng. 2017. Bacterial and
Fungal Community Structures in Loess Plateau Grasslands with Different
Grazing Intensities. *Frontiers in Microbiology* 8:606. doi:
10.3389/fmicb.2017.00606

Hymer, W. C., and W. J. Kraemer. 2023. Resistance exercise stress:
theoretical mechanisms for growth hormone processing and release from the
anterior pituitary somatotroph. *European journal of applied physiology*
123(9):1867-1878. doi: 10.1007/s00421023052638

Jing, X., L. Ding, J. Zhou, X. Huang, A. Degen, and R. Long. 2022. The
adaptive strategies of yaks to live in the Asian highlands. *Animal Nutrition*
9:249-258. doi: <https://doi.org/10.1016/j.aninu.2022.02.002>

Krishnan, G., V. Paul, S. Hanah, J. Bam, and P. Das. 2016. Effects of
climate change on yak production at high altitude. *The Indian journal of*
*animal sciences* 86:621-626. doi: 10.56093/ijans.v86i6.59033

Li, A., C. Liu, X. Han, J. Zheng, G. Zhang, X. Qi, P. Du, and L. Liu.
2023a. Tibetan Plateau yak milk: A comprehensive review of nutritional
values, health benefits, and processing technology. *Food Chemistry: X*
20:100919. doi: <https://doi.org/10.1016/j.fochx.2023.100919>

Li, D., S. Xu, H. Jiang, Y. T. Li, Y. Zhao, P. Jin, S. Ji, Y. Du, and D. Q.
Tang. 2024. Gut microbiota and intestinal FXR signaling involved in the
alleviation of mulberry (*Morus alba* L.) leaf ethanol extract on type 2 diabetes
mellitus in db/db mice. *Journal of Functional Foods* 123:106600. doi:
<https://doi.org/10.1016/j.jff.2024.106600>

Li, W., L. Yue, and S. Xiao. 2021. Increase in Right Temporal Cortex
Thickness Is Related to Decline of Overall Cognitive Function in Patients
With Hypertension. *Frontiers in Cardiovascular Medicine* 8:758787. doi:
10.3389/fcvm.2021.758787

Li, Z., H. Qu, L. Li, J. Zheng, D. Wei, and F. Wang. 2023b. Effects of
climate change on vegetation dynamics of the Qinghai-Tibet Plateau, a
causality analysis using empirical dynamic modeling. *Heliyon* 9(5):e16001.
doi: 10.1016/j.heliyon.2023.e16001

Liu, X., J. Gao, S. Liu, Y. Cheng, L. Hao, S. Liu, and W. Zhu. 2023. The
uniqueness and superiority of energy utilization in yaks compared with cattle
in the highlands: A review. *Animal Nutrition* 12:138-144. doi:
10.1016/j.aninu.2022.09.011

Liu, Y., X. Meng, C. Tang, L. Zheng, K. Tao, and W. Guo. 2024.
Aerobic exercise modulates RIPK1-mediated MAP3K5/JNK and NF- κ B
pathways to suppress microglia activation and neuroinflammation in the
hippocampus of D-gal-induced accelerated aging mice. *Physiology &*
*Behavior* 286:114676. doi: <https://doi.org/10.1016/j.physbeh.2024.114676>

Long, R. J., S. O. Apori, F. B. Castro, and E. R. Ørskov. 1999. Feed
value of native forages of the Tibetan Plateau of China. *Animal Feed Science*
and *Technology* 80(2):101-113. doi:
[https://doi.org/10.1016/S0377-8401\(99\)00057-7](https://doi.org/10.1016/S0377-8401(99)00057-7)

Lu, M., J. U. Flanagan, R. J. Langley, M. P. Hay, and J. K. Perry. 2019.
Targeting growth hormone function: strategies and therapeutic applications.
*Signal Transduction and Targeted Therapy* 4:3. doi: 10.1038/s413920190036y

Luo, B., D. Xiang, X. Ji, X. Chen, R. Li, S. Zhang, Y. Meng, D. C.
Nieman, and P. Chen. 2024. The anti-inflammatory effects of exercise on
autoimmune diseases: A 20-year systematic review. *Journal of Sport and*
*Health Science* 13(3):353-367. doi: <https://doi.org/10.1016/j.jshs.2024.02.002>

548 Ma, L., S. Xu, H. Liu, T. Xu, L. Hu, N. Zhao, X. Han, and X. Zhang.
2019. Yak rumen microbial diversity at different forage growth stages of an
alpine meadow on the Qinghai-Tibet Plateau. *PeerJ* doi: 10.7717/peerj.7645

Mao, X., R. Sun, Q. Wang, D. Chen, B. Yu, J. He, J. Yu, J. Luo, Y. Luo,
H. Yan, J. Wang, H. Wang, and Q. Wang. 2021. l-Isoleucine Administration
Alleviates DSS-Induced Colitis by Regulating TLR4/MyD88/NF-κB Pathway
in Rats. *Frontiers in Immunology* 12:817583. doi:
10.3389/fimmu.2021.817583

Mipam, T. D., L. L. Zhong, J. Q. Liu, G. Mieke, and L. M. Tian. 2019.
Productive Overcompensation of Alpine Meadows in Response to Yak
Grazing in the Eastern Qinghai-Tibet Plateau. *Frontiers in Plant Science*
10(Original Research) doi: 10.3389/fpls.2019.00925

Nijenhuis-Noort, E. C., K. A. Berk, S. J. C. M. M. Neggers, and A. J. v. d.
Lely. 2024. The Fascinating Interplay between Growth Hormone, Insulin-Like
Growth Factor-1, and Insulin. *Endocrinology and metabolism (Seoul, Korea)*
39(1):83-89. doi: 10.3803/EnM.2024.101

Okada, S., and J. J. Kopchick. 2001. Biological effects of growth
hormone and its antagonist. *Trends in Molecular Medicine* 7(3):126-132. doi:
[https://doi.org/10.1016/S14714914\(01\)019335](https://doi.org/10.1016/S14714914(01)019335)

Powers, S. K., E. R. Goldstein, M. A. Schragar, and L. L. Ji. 2023.
Exercise Training and Skeletal Muscle Antioxidant Enzymes: An Update.
*Antioxidants* 12(1):39.

Qiu, Q., G. Zhang, T. Ma, W. Qian, J. Wang, Z. Ye, C. Cao, Q. Hu, J.
Kim, D. M. Larkin, L. Auvil, B. Capitanu, J. Ma, H. A. Lewin, X. Qian, Y.
Lang, R. Zhou, L. Wang, K. Wang, J. Xia, S. Liao, S. Pan, X. Lu, H. Hou, Y.
Wang, X. Zang, Y. Yin, H. Ma, J. Zhang, Z. Wang, Y. Zhang, D. Zhang, T.
Yonezawa, M. Hasegawa, Y. Zhong, W. Liu, Y. Zhang, Z. Huang, S. Zhang,
R. Long, H. Yang, J. Wang, J. A. Lenstra, D. N. Cooper, and Y. Wu. 2012.
The yak genome and adaptation to life at high altitude. *Nature Genetics*
44(8):946-949. doi: 10.1038/ng.2343

Sabag, A., D. Chang, and N. A. Johnson. 2021. Growth Hormone as a
Potential Mediator of Aerobic Exercise-Induced Reductions in Visceral
Adipose Tissue. *Frontiers in Physiology* 12. doi: 10.3389/fphys.2021.623570

Sapkota, S., K. P. Acharya, R. Laven, and N. A. 4. 2022. Possible
Consequences of Climate Change on Survival, Productivity and Reproductive
Performance, and Welfare of Himalayan Yak (*Bos grunniens*). *Veterinary*
*Sciences* 9(8)doi: 10.3390/vetsci9080449

Shah, A. M., I. Bano, I. H. Qazi, M. Matra, and M. Wanapat. 2023. “The
Yak”—A remarkable animal living in a harsh environment: An overview of its
feeding, growth, production performance, and contribution to food security.
*Frontiers in Veterinary Science* doi: 10.3389/fvets.2023.1086985

Stanley, T. L., and S. K. Grinspoon. 2015. Effects of growth
hormone-releasing hormone on visceral fat, metabolic, and cardiovascular
indices in human studies. *Growth Hormone & IGF Research* 25(2):59-65. doi:
10.1016/j.ghir.2014.12.005

Syu, G. D., H. I. Chen, and C. J. Jen. 2012. Differential effects of acute
and chronic exercise on human neutrophil functions. *Medicine and Science in*
*Sports and Exercise* 44(6):1021-1027. doi: 10.1249/MSS.0b013e3182408639

Vaintrub, M. O., H. Levit, M. Chincarini, I. Fusaro, M. Giammarco, and
G. Vignola. 2021. Review: Precision livestock farming, automats and new
technologies: possible applications in extensive dairy sheep farming. *Animal*
15(3):100143. doi: 10.1016/j.animal.2020.100143

Wang, G., X. Zhao, J. Zhong, M. Cao, Q. He, Z. Liu, Y. Lin, Y. Xu, and
Y. Zheng. 2013. Cloning and polymorphisms of yak lactate dehydrogenase B
gene. *Int J Mol Sci* 14(6):11994-12003. doi: 10.3390/ijms140611994

Wang, H., Y. Wang, L. Yang, J. Feng, S. F. Tian, L. Chen, W. Huang, J.
Liu, and X. Wang. 2024a. Integrated 16S rRNA sequencing and
metagenomics insights into microbial dysbiosis and distinct virulence factors
in inflammatory bowel disease. *Frontiers in Microbiology* 15.doi:
10.3389/fmicb.2024.1375804

Wang, X., T. Guo, Q. Zhang, N. Zhao, L. Hu, H. Liu, and S. Xu. 2024b.
Seasonal variations in composition and function of gut microbiota in grazing
yaks: Implications for adaptation to dietary shift on the Qinghai-Tibet plateau.
*Ecology and Evolution* 14(10):e70337. doi:
<https://doi.org/10.1002/ece3.70337>

Wu, Y., B. Wang, L. Tang, Y. Zhou, Q. Wang, L. Gong, J. Ni, and W. Li.
2022. Probiotic *Bacillus* Alleviates Oxidative Stress-Induced Liver Injury by
Modulating Gut-Liver Axis in a Rat Model. *Antioxidants (Basel)* 11(2)doi:
10.3390/antiox11020291

Xin, J., Z. Chai, C. Zhang, Q. Zhang, Y. Zhu, H. Cao, J. Zhong, and Q. Ji.
2019. Comparing the Microbial Community in Four Stomach of Dairy Cattle,
Yellow Cattle and Three Yak Herds in Qinghai-Tibetan Plateau. *Frontiers in*
*Microbiology* 10:1547. doi: 10.3389/fmicb.2019.01547

Xue, B., J. X. Zhang, Z. S. Wang, L. Z. Wang, Q. Peng, L. Da, S. Bao,
and X. Kong. 2021. Metabolism response of grazing yak to dietary concentrate
supplementation in warm season. *Animal* 15(3):100175. doi:
<https://doi.org/10.1016/j.animal.2021.100175>

Yang, S., J. Zheng, H. Mao, P. Vinitchaikul, D. Wu, and J. Chai. 2024a.
Multiomics of yaks reveals significant contribution of microbiome into host
metabolism. *npj Biofilms and Microbiomes* 10(1):133. doi:
10.1038/s41522024006092

Yang, X., U. Daraz, J. Ma, X. Lu, Q. Feng, H. Zhu, and X. Wang. 2024b.
Temporal-spatial variability of grazing behaviors of yaks and the drivers of
their intake on the eastern Qinghai-Tibetan Plateau. *Frontiers in Veterinary*
*Science* doi: 10.3389/fvets.2024.1393136

Yi, S., H. Wu, Y. Liu, D. Dai, Q. Meng, S. Chai, S. Liu, and Z. Zhou.
2023. Concentrate supplementation improves cold-season environmental
fitness of grazing yaks: responsive changes in the rumen microbiota and
metabolome. *Frontiers in Microbiology* 14:1247251. doi:
10.3389/fmicb.2023.1247251

Yu, M., C. Zhang, Y. Yang, C. Mu, Y. Su, K. Yu, and W. Zhu. 2017.
Long-term effects of early antibiotic intervention on blood parameters,
apparent nutrient digestibility, and fecal microbial fermentation profile in pigs
with different dietary protein levels. *Journal of Animal Science and*
*Biotechnology* 8:60. doi: 10.1186/s4010401701922

Zhang, B., X. Wang, Z. Ding, Y. Kang, S. Guo, M. Cao, L. Hu, L. Xiong,
644 J. Pei, and X. Guo. 2024. Effects of High-Concentrate Diets on Growth
Performance, Serum Biochemical Indexes, and Rumen Microbiota in
House-Fed Yaks. *Animals* 14(24):3594.

Zhang, Y., M. Wang, X. Wang, R. Li, R. Zhang, W. Xun, H. Li, X. Xin,
and R. Yan. 2023. Grazing Regulates Changes in Soil Microbial Communities
in Plant-Soil Systems. *Agronomy* 13(3):708.

Zhao, F., C. Ren, S. Shelton, Z. Wang, G. Pang, J. Chen, and J. Wang.
2017. Grazing intensity influence soil microbial communities and their
implications for soil respiration. *Agriculture, Ecosystems & Environment* doi:
10.1016/j.agee.2017.08.007

Zheng, Z., Y. Zong, Y. Ma, Y. Tian, Y. Pang, C. Zhang, and J. Gao. 2024.
Glucagon-like peptide-1 receptor: mechanisms and advances in therapy.
*Signal Transduction and Targeted Therapy* 9(1):234. doi:
10.1038/s4139202401931z

Zhu, Y., X. Li, L. zhaxi, S. zhaxi, Suolang, Ciyang, G. Sun, C. yangj, and
B. wangdu. 2022. House feeding system improves the estrus rate in yaks (*Bos*
*grunniens*) by increasing specific fecal microbiota and myo-inositol content in
serum. *Frontiers in Microbiology* 13:974765. doi: 10.3389/fmicb.2022.974765

Zhu, Y., J. Wang, Y. Cidan, H. Wang, K. Li, and W. Basang. 2024. Gut
Microbial Adaptation to Varied Altitudes and Temperatures in Tibetan Plateau
Yaks. *Microorganisms* 12(7):1350.

Zou, F., Q. Hu, H. Li, J. Lin, Y. Liu, and F. Sun. 2021. Dynamic
Disturbance Analysis of Grasslands Using Neural Networks and
Spatiotemporal Indices Fusion on the Qinghai-Tibet Plateau. *Frontiers in Plant*
*Science* 12:760551. doi: 10.3389/fpls.2021.760551

Table 1 Ingredients and nutritional level of experiment diets.

Diet composition	Content (% of DM)	Nutritional level ²	
Alfalfa hay	16.66	CP, %	11.62
Oat hay	16.66	EE, %	4.06
Full corn silage	20	NDF, %	31.69
Corn	38.50	ADF, %	18.80
Rapeseed oil	0.5	ADL, %	3.56
Wheat bran	5.00	NFC, %	48.32
Soybean meal	1.50	Starch, %	28.2
Cottonseed meal	1.00	Ash, %	4.31
Rapeseed meal	1.00	Ca, %	0.86
Calcium carbonate	1.58	P, %	0.41
Calcium hydrogen phosphate	0.44	ME, MJ/kg DM ³	7.82
Premix ¹	1.00		

¹The premix provides per kg diet: 10mg Cu in the form of sulfate, 60mg Zn in the
form of sulfate, 50mg Mn in the form of sulfate, 50mg Fe in the form of sulfate,
Co in the form of chloride 0.2mg, I in the form of iodate 0.5mg, Se in the form of
selenite 0.3mg, and Vitamin A 10,000 IU, Vitamin D₃ 2,000 IU, and Vitamin E 60 IU.

²The ME value of TMR was calculated based on the available ME data of the ingredients.
CP, EE, NDF, ADF, ADL, Ca, and P were determined values.
NFC=DM-(EE%+CP%+CP%+ASH%).DM, dry matter; ME, metabolizable energy; CP, crude
protein; NDF, neutral detergent fiber; NFC, non-fiber carbohydrate; ADF, acid
detergent fiber; ADL, acid detergent lignin; Ca, calcium; P, phosphorus.

Table 2 Effect of Feeding Patterns on the Growth Performance of Yak

Items	Feeding Mode ¹ (Mean \pm SD)		P-value
	Grazing yak	Housing yak	
Initial Body Weight, kg	55.79 \pm 4.52	56.16 \pm 4.35	0.6053
Final Body Weight, kg	75.95 \pm 4.70	72.21 \pm 4.81	0.0389
Average Daily Gain, g/d	223.98 \pm 27.55	178.36 \pm 26.99	<0.0001
Average Daily Feed Intake, kg/d		5.51 \pm 0.069	

¹ Grazing yak: Grazing yak group (n=18); Housing yak: Housing yak group (n=18). The
same applies to the following tables.

Table 3: Effect of Feeding Patterns on Serum Biochemical Indicators of Yak

Items	Feeding Mode (Mean \pm SD)		P-value
	Grazing yak	Housing yak	
Total Protein, g/L	70.77 \pm 7.14	70.95 \pm 6.79	0.9370
Albumin, g/L	43.34 \pm 5.90	40.55 \pm 4.44	0.1084
Globulin, g/L	27.44 \pm 4.64	30.40 \pm 4.17	0.0450
Blood Urea Nitrogen, mmol/L	3.27 \pm 0.63	6.12 \pm 1.35	<0.0001
Total Cholesterol, mmol/L	3.09 \pm 0.53	3.51 \pm 1.10	0.3349
Triglyceride, mmol/L	1.57 \pm 0.75	1.37 \pm 0.20	0.4214
Glucose, mmol/L	5.25 \pm 0.73	4.80 \pm 0.87	0.3179
Alanine Aminotransferase, U/L	98.36 \pm 26.11	102.41 \pm 16.08	0.5682
Aspartate Aminotransferase, U/L	45.60 \pm 5.30	36.43 \pm 5.90	0.0398
Alkaline Phosphatase, U/L	156.05 \pm 17.17	163.25 \pm 23.70	0.6934

Table 4 Effects of Different Feeding Patterns on Serum Antioxidant Indicators in
Yaks

Items	Feeding Mode (Mean \pm SD)		P-value
	Grazing yak	Housing yak	
Total Superoxide Dismutase, U/mL	470.26 \pm 32.00	385.29 \pm 23.37	<0.0001
Total Antioxidant Capacity, U/mL	49.46 \pm 2.62	46.61 \pm 2.78	0.0024
Malondialdehyde, nmol/ml	209.99 \pm	184.43 \pm	0.6954

		29.65	36.55	
Glutathione Peroxidase, U/mL	9.02 ± 1.63	8.66 ± 1.21	0.3190

Table 5: Effects of Different Feeding Patterns on Serum Immune Indicators in Yaks

Items	Feeding Mode (Mean \pm SD)		P -value
	Grazing yak	Housing yak	
Immunoglobulin A, g/L	1.64 \pm 0.05	1.32 \pm 0.07	<0.0001
Immunoglobulin M, g/L	1.37 \pm 0.05	1.13 \pm 0.05	<0.0001
Immunoglobulin G, g/L	7.52 \pm 0.34	6.05 \pm 0.30	<0.0001
Interleukin 2, pg/mL	426.18 \pm 32.42	314.91 \pm 32.25	<0.0001
Interleukin 4, pg/mL	29.85 \pm 5.24	49.89 \pm 5.45	<0.0001
Interleukin 6, pg/mL	281.97 \pm 22.74	199.05 \pm 24.86	<0.0001
Interleukin 10, pg/mL	93.94 \pm 21.26	161.83 \pm 20.32	<0.0001
Tumor Necrosis Factor α , pg/mL	261.41 \pm 21.69	193.51 \pm 20.56	<0.0001
Interferon γ , pg/mL	2147.50 \pm 182.88	1564.69 \pm 146.79	<0.0001

Table 6: Effects of Different Feeding Patterns on Serum Growth Hormone Axis
Indicators in Yaks

Items	Feeding Mode (Mean \pm SD)		P -value
	Grazing yak	Housing yak	
Growth Hormone, ng/mL	15.93 \pm 1.09	10.51 \pm 1.27	<0.0001
Insulin-like Growth Factor 1, ng/mL	389.21 \pm 26.89	249.67 \pm 29.93	<0.0001
Growth Hormone Releasing Hormone, pg/mL	37.38 \pm 1.99	26.34 \pm 1.77	<0.0001
Growth Hormone Inhibiting Hormone, pg/mL	203.90 \pm 18.05	306.30 \pm 16.12	<0.0001

Fig. 1 Plasma metabolome analysis diagrams. (A) PCA score plot; (B) PLS - DA score
 plot; (C) Volcano plot of differential metabolites; (D) KEGG enrichment plot

Fig. 2 Alpha diversity, beta diversity, and microbiota composition at phylum and genus levels of fecal microbiota in different treatment groups. (A) ACE index. (B) *Chao1* index. (C) Shannon index. (D) Beta diversity. (E) Microbiota at phylum level; (F) Microbiota at genus

level.

Fig. 3 LEfSe multi-level species differential discriminant analysis. (A) LEfSe multi-level species hierarchical tree diagram; (B) LDA

discrimination result table.

Fig. 4 Radar charts, KEGG enrichment diagrams, and Mantel-test Network heatmaps of fecal short-chain fatty acid of feces in different treatment groups. (A) Radar chart; (B) KEGG enrichment diagrams; (C) Mantel-test Network heatmaps.

This current research is to investigate the effects of two feeding patterns including a warm-season grazing and a pen-feeding pattern on yaks. The parameters of growth performance, antioxidant capacity, immune function, metabolome, and fecal microbile composition were detected to assess the differences between two feeding treatments. The experiment is well-designed and rigorously conducted. The manuscript is well-written and logically structured.

However, it's not convincing to draw the conclusion that grazing pattern (pasture-based system) is better than pen-feeding system (generally called intensive system) based on the current data. You can say the detected parameters are better for yaks raised under grazing system than pen-feeding system in current experiment, but it's very dangerous to say grazing system is better. For instance, if the nutrition level, the living condition, or the management are optimized under the intensive system, the results might be different. I recommend to modify this statement throughout the context.

The practical or estimated feed intake data under grazing system is needed.

Other comments are listed as follows.

L27, found is not the right word.

L57 correct this sentence: these practices are designed to optimize yak growth rates, improve feed utilization, and reduce environmental disturbances.

L87 change "and" to "with"

L84 the ethic statement for conducting this animal trial is mandatory.

L91 the pictures of grazing and pen-feeding conditions would help readers to understand the differences of two systems

L91 add country name.

L92 add space for 30°08'N

L92 modify the animal amount.

L97 one yak

L105 why the samples were kept under dark?

L106 list the number of fecal samples

L186 did all data are analyzed on this platform? Otherwise, supplement how the data of serum, plasma parameters, and growth data were analyzed.

L207 the form of $P < 0.05$ need to be unified throughout the text. P or p. with spaces or not?

L221-231 the words are in different forms.

L247 How did you collect 38 samples from 36 animals? The quality control samples should be at least 3 samples.

L292 intake to ingest

L295 space is needed after yaks. This happens a lot in this text. Check them all.

L296 exercises

L324 cite references to support this statement of more activities will stimulate the hormone secretion.

L400, L401, modify the statement of grazing pattern is better..... than pen-fed system.

L426 modify the reference form.

L670 unify the form of all table titles, and the animal, sample, treatment information are needed in the titles of tables and figures.

ME, MJ/kg DM³

L677 modify the formula

Dear Editor,

Thank you very much for your and the reviewers' valuable comments and suggestions. Based on your feedback, we have carefully revised the manuscript as follows:

Reviewer 2 left me a message suggesting that the authors further clarify the following points. Please respond as well.

1. Statistical Methods: A more detailed explanation of the statistical methods used, including how the sample size was determined, would strengthen the rigor of the analysis and ensure the robustness of the results.

Response: We have provided a more detailed explanation of the statistical methods used in our study. Specifically, we have elaborated on the sample size determination and the rationale behind our statistical approach. In the Methods section, we have clearly stated the number of replicates for each sample (lines 97, 106, and 18), which supports the robustness of our results. Additionally, we have included a detailed description of the statistical tests employed and the assumptions made to ensure the rigor of our analysis.

2. Treatment Clarification: It would be important to confirm whether the yaks received any treatments (e.g., antibiotics, anthelmintics, vaccines) during the trial, as these could potentially affect the gut microbiota results and overall interpretation of the findings.

Response: We have explicitly stated that none of the yaks received any antibiotics, vaccines, or other treatments during the trial (lines 109-110). This ensures that our findings on gut microbiota are not confounded by external treatments and provides a more accurate interpretation of our results.

3. Sequencing Data Quality: I recommend that the authors confirm the consistency and quality of the sequencing data, and include a more detailed description of sequencing depth, OTUs operational taxonomic units count, and detection limits in the methods section to enhance transparency.

Response: We have included a more detailed description of the sequencing data quality in the Methods section. Specifically, we have provided information on the sequencing depth, operational taxonomic units (OTUs) count, and detection limits. Additionally, we have also summarized the key details of the sequencing data in the Results section (lines 256-262) to ensure that readers can easily access this important information.

Reviewer #1

1. The author should carefully check the Latin terms in the manuscript, which should be italicized, such as "*Bos grunniens*" (line 45).

Response: We have carefully reviewed the manuscript and have italicized the Latin term "*Bos grunniens*" at the specified locations (lines 24 and 54). This change has been made to adhere to the standard scientific formatting conventions.

2. Consistency of Terminology: The author needs to standardize the use of specialized terms in the text. For example, expressions such as "penned fed," "Housing yak," and "pen-feeding" should be consistent to avoid confusion.

Response: We agree that standardizing specialized terms is crucial for clarity and coherence. In response to your comment, we have thoroughly reviewed the manuscript and standardized the use of terms related to the feeding and housing conditions of the yaks. Specifically, we have replaced

inconsistent expressions such as "penned fed," "Housing yak," and "pen-feeding" with the consistent term "housing yak" throughout the text.

3. At line 94, "yaks" should be in the singular form.

Response: In response to your comment at line 94, we have reviewed the usage of the term "yaks" and have made the necessary revisions to ensure grammatical accuracy.

We have changed "yaks" to the singular form "yak" at line 94, and we have also reviewed the entire manuscript to ensure consistent and correct usage of the singular and plural forms of "yak" throughout the text. This change aligns with the suggestions from both reviewers and enhances the clarity and coherence of our manuscript.

4. Microorganisms at the phylum level do not need to be italicized, such as "Actinobacteriota" at line 250 and "Bacteroidota" at line 251. The author should also carefully check the rest of the manuscript for consistency.

Response: We have carefully reviewed the manuscript and made the necessary corrections to ensure consistency and adherence to scientific conventions. In response to your comment, we have corrected the formatting of "Actinobacteriota" and "Bacteroidota" at line 267. These terms are now presented in regular typeface, as they should not be italicized at the phylum level. Additionally, we have conducted a thorough check of the entire manuscript to ensure that all phylum-level names are consistently formatted and not italicized.

5. At line 192, " $p < 0.05$ " should be capitalized as " $P < 0.05$ ".

Response: We have capitalized " p " to " P " at line 203, ensuring that it is consistent with the standard convention of capitalizing the " P " in " $P < 0.05$ " throughout the manuscript. This change enhances the accuracy and readability of our statistical reporting.

6. The author should refer to the journal's format and revise the citation style of the manuscript.

Response: We have updated the citations throughout the manuscript to conform to the journal's required format. The changes have been made in the reference section (lines 443-665) to ensure consistency and adherence to the journal's standards.

Reviewer #2 (Comments for the Author)

Reviewer #2 (Comments for the Author):

The study titled "Effects of Warm-Season Feeding Patterns on Growth Performance, Antioxidant Capacity, Immune Function, Metabolome, and Fecal Microbiota in Yaks" presents a well conducted and insightful analysis. It effectively integrates growth performance, immune function, metabolomics, and gut microbiota, making it highly relevant to yak husbandry and sustainable livestock production under climate change. However I suggest the following improvements to enhance clarity, scientific rigor, and readability follows as:

1. Title and Abstract: The title is informative but slightly lengthy. Consider simplifying it by grouping related terms or focusing on the key outcomes. The abstract is comprehensive but dense. Breaking up long sentences and reducing redundancy would enhance readability. Ensure *Bos grunniens* is italicized. Use the term "yaks" instead of "yak" for consistency and grammatical correctness.

Response:

Title, we have revised the title to make it more concise while retaining its informative nature. The new title is now more streamlined and focuses on the key outcomes of our study (revised in lines 1-4).

Abstract, We have revised the abstract to improve its readability. We have broken up long sentences and reduced redundancy to make the content more accessible. The revised abstract is now clearer and more concise (revised in lines 24-51).

Regarding the use of the term "yak" versus "yaks," we have reviewed the manuscript and have standardized the terminology for consistency and grammatical correctness. We have used "yak" in the singular form throughout the manuscript to maintain uniformity.

Additionally, we have ensured that "*Bos grunniens*" is italicized in the abstract and throughout the manuscript to adhere to scientific conventions.

2.Introduction: The introduction provides a well-rounded background on yak biology, ecological importance, and evolving husbandry practices. The rationale for comparing grazing and pen-feeding systems is presented. The hypothesis is logical but could be rephrased in a more concise and testable format. For example: "We hypothesize that warm-season grazing improves metabolic efficiency and immune responses compared to pen-feeding." Consider including one or two guiding research questions, such as: How do grazing and pen feeding differ in terms of long-term sustainability for yak populations, particularly under the pressure of climate change? Do shifts in gut microbiota in grazing yaks enhance metabolism or disease resistance?

Response: In response to your feedback, we have revised the Introduction to present the hypothesis in a more concise and testable format. We have also included guiding research questions to provide a clearer direction for our study. The revised hypothesis and research questions are now presented in lines 80-85

3.Materials and Methods. Please clarify whether the yaks received any antibiotic, anthelmintic, or vaccine treatments before or during the trial. This is critical for interpreting the microbiome and metabolomics data. Justify the selection of the V3-V4 region for 16S rRNA sequencing. Include basic sequencing statistics such as sequencing depth, OTUs count, and coverage to enhance transparency. The extraction and analysis methods are sound; however, including detection limits and rationale for the choice of solvents would increase methodological rigor.

Response: We have carefully considered your suggestions and have made the following revisions and clarifications:

Treatments for Yaks: We have explicitly stated in the manuscript that no antibiotics, anthelmintics, or vaccines were administered to the yaks before or during the trial. This information is clearly mentioned in the Methods section (lines 109-110) to ensure transparency and to support the interpretation of our microbiome and metabolomics data.

In addition, the V3-V4 region was chosen because it offers several advantages for microbial studies, including high species resolution, moderate sequence length, effective universal primer design, comprehensive database support, and good data comparability. We have included detailed sequencing statistics to enhance transparency. The sequencing depth, OTUs count, and coverage are described in both the Methods section (lines 158-170 under "Fecal microbiota") and the Results section (lines 252-258 under "Fecal

microbiota and short-chain fatty acids").

4. Results The results are structured. Including descriptive statistics (e.g., mean \pm SD) alongside p-values or significance indicators will improve interpretability. Try linking specific biomarker changes (e.g. immune cytokines or metabolites) to physiological processes such as liver function, nutrient metabolism, or muscle activity.

We have carefully considered your suggestions and have made the following revisions to enhance the clarity and interpretability of our results:

Descriptive Statistics and Significance Indicators: To improve the interpretability of our results, we have included descriptive statistics in the form of mean \pm SD alongside p-values or significance indicators. This information is now clearly presented in the tables, allowing readers to better understand the magnitude and significance of the observed changes.

Linking Biomarker Changes to Physiological Processes: We have strengthened the connection between specific biomarker changes (e.g., immune cytokines or metabolites) and physiological processes such as liver function, nutrient metabolism, or muscle activity. While the primary presentation of these links is in the Discussion section, we have also ensured that the results are framed in a way that highlights their relevance to these physiological processes.

5. Discussion: The discussion is comprehensive and well linked to the study objectives. Consider being more concise by focusing on the key findings and reducing redundancy, especially in sections about hormone regulation and immune responses. The integration of microbiota and metabolomics is a major strength this section would benefit from streamlining to highlight the novelty and significance of these findings.

We have carefully considered your suggestions and have revised the section to enhance its conciseness and focus on the key findings.

In response to your comments, we have streamlined the content related to hormone regulation and immune responses, reducing redundancy and focusing on the most significant results. This revision helps to highlight the novelty and importance of our findings, particularly in the integration of microbiota and metabolomics. The changes have been made in lines 304-351 to ensure that the discussion is more concise and impactful.

We believe these revisions improve the clarity and relevance of the Discussion section. Thank you again for your insightful comments.

6. Conclusion. The conclusion is generally satisfactory. It would be stronger if it briefly mentioned the underlying mechanisms and offered clear future research directions based on the study findings.

We have carefully considered your suggestions and have made the following revisions to enhance the strength and clarity of our conclusion.

In response to your comments, we have included a brief discussion of the underlying mechanisms observed in our study, as well as clear future research directions based on our findings. This additional information has been incorporated into the Conclusion section (lines 411-416).

We believe these changes provide a more comprehensive and forward-looking conclusion, highlighting the significance of our study and suggesting potential avenues for further research.

Reviewer #2 (Comments for the Author):

The study titled "Effects of Warm-Season Feeding Patterns on Growth Performance, Antioxidant Capacity, Immune Function, Metabolome, and Fecal Microbiota in Yaks" presents a well conducted and insightful analysis. It effectively integrates growth performance, immune function, metabolomics, and gut microbiota, making it highly relevant to yak husbandry and sustainable livestock production under climate change. However I suggest the following improvements to enhance clarity, scientific rigor, and readability follows as:

1. Title and Abstract: The title is informative but slightly lengthy. Consider simplifying it by grouping related terms or focusing on the key outcomes. The abstract is comprehensive but dense. Breaking up long sentences and reducing redundancy would enhance readability. Ensure *Bos grunniens* is italicized. Use the term "yaks" instead of "yak" for consistency and grammatical correctness.

Response:

Title, we have revised the title to make it more concise while retaining its informative nature. The new title is now more streamlined and focuses on the key outcomes of our study (revised in lines 1-4).

Abstract, we have revised the abstract to improve its readability. We have broken up long sentences and reduced redundancy to make the content more accessible. The revised abstract is now clearer and more concise (revised in lines 24-51).

Regarding the use of the term "yak" versus "yaks," we have reviewed the manuscript and have standardized the terminology for consistency and grammatical correctness. We have used "yak" in the singular form throughout the manuscript to maintain uniformity.

Additionally, we have ensured that "*Bos grunniens*" is italicized in the abstract and throughout the manuscript to adhere to scientific conventions.

2. Introduction: The introduction provides a well-rounded background on yak biology, ecological importance, and evolving husbandry practices. The rationale for comparing grazing and pen-feeding systems is presented. The hypothesis is logical but could be rephrased in a more concise and testable format. For example: "We hypothesize that warm-season grazing improves metabolic efficiency and immune responses compared to pen-feeding." Consider including one or two guiding research questions, such as: How do grazing and pen feeding differ in terms of long-term sustainability for yak populations, particularly under the pressure of climate change? Do shifts in gut microbiota in grazing yaks enhance metabolism or disease resistance?

Response: In response to your feedback, we have revised the Introduction to present the hypothesis in a more concise and testable format. We have also included guiding research questions to provide a clearer direction for our study. The revised hypothesis and research questions are now presented in lines 80-85

3. Materials and Methods. Please clarify whether the yaks received any antibiotic, anthelmintic, or vaccine treatments before or during the trial. This is critical for interpreting the microbiome and metabolomics data. Justify the selection of the V3-V4 region for 16S rRNA sequencing. Include basic sequencing statistics such as sequencing depth, OTUs count, and coverage to enhance transparency. The extraction and analysis methods are sound; however, including detection limits and rationale for the choice of solvents would increase methodological rigor.

Response: We have carefully considered your suggestions and have

made the following revisions and clarifications:

Treatments for Yaks: We have explicitly stated in the manuscript that no antibiotics, anthelmintics, or vaccines were administered to the yaks before or during the trial. This information is clearly mentioned in the Methods section (lines 109-110) to ensure transparency and to support the interpretation of our microbiome and metabolomics data.

In addition, the V3-V4 region was chosen because it offers several advantages for microbial studies, including high species resolution, moderate sequence length, effective universal primer design, comprehensive database support, and good data comparability. We have included detailed sequencing statistics to enhance transparency. The sequencing depth, OTUs count, and coverage are described in both the Methods section (lines 158-170 under "Fecal microbiota") and the Results section (lines 252-258 under "Fecal microbiota and short-chain fatty acids").

4.Results The results are structured. Including descriptive statistics (e.g., mean \pm SD) alongside p-values or significance indicators will improve interpretability. Try linking specific biomarker changes (e.g immune cytokines or metabolites) to physiological processes such as liver function, nutrient metabolism, or muscle activity.

We have carefully considered your suggestions and have made the following revisions to enhance the clarity and interpretability of our results:

Descriptive Statistics and Significance Indicators: To improve the interpretability of our results, we have included descriptive statistics in the form of mean \pm SD alongside p-values or significance indicators. This information is now clearly presented in the tables, allowing readers to better understand the magnitude and significance of the observed changes.

Linking Biomarker Changes to Physiological Processes: We have strengthened the connection between specific biomarker changes (e.g., immune cytokines or metabolites) and physiological processes such as liver function, nutrient metabolism, or muscle activity. While the primary presentation of these links is in the Discussion section, we have also ensured that the results are framed in a way that highlights their relevance to these physiological processes.

5. Discussion: The discussion is comprehensive and well linked to the study objectives. Consider being more concise by focusing on the key findings and reducing redundancy, especially in sections about hormone regulation and immune responses. The integration of microbiota and metabolomics is a major strength this section would benefit from streamlining to highlight the novelty and significance of these findings.

We have carefully considered your suggestions and have revised the section to enhance its conciseness and focus on the key findings.

In response to your comments, we have streamlined the content related to hormone regulation and immune responses, reducing redundancy and focusing on the most significant results. This revision helps to highlight the novelty and importance of our findings, particularly in the integration of microbiota and metabolomics. The changes have been made in lines 304-351 to ensure that the discussion is more concise and impactful.

We believe these revisions improve the clarity and relevance of the Discussion section. Thank you again for your insightful comments.

6.Conclusion. The conclusion is generally satisfactory. It would be

stronger if it briefly mentioned the underlying mechanisms and offered clear future research directions based on the study findings.

We have carefully considered your suggestions and have made the following revisions to enhance the strength and clarity of our conclusion.

In response to your comments, we have included a brief discussion of the underlying mechanisms observed in our study, as well as clear future research directions based on our findings. This additional information has been incorporated into the Conclusion section (lines 411-416).

We believe these changes provide a more comprehensive and forward-looking conclusion, highlighting the significance of our study and suggesting potential avenues for further research.

7. Data availability: The data availability statement should be more specific. Please include an accession number (e.g. NCBI SRA) or provide a link to an online repository if available for better transparency.

We fully agree that providing specific details about the data repository enhances transparency and reproducibility.

In response to your suggestion, we have updated the Data Availability statement to include the accession number and a direct link to the NCBI SRA repository where our data are stored. This information can be found in lines 435-438.

Thank you very much for your time and effort in reviewing our manuscript. We truly appreciate the constructive comments and suggestions provided by you and the reviewers, which have been instrumental in helping us improve the quality of our work. In response to your feedback, we have made a series of revisions to enhance the clarity, accuracy, and overall presentation of our manuscript. We believe these changes have significantly strengthened our study and addressed the concerns raised.

We hope that these revisions meet your expectations and that you will find our manuscript suitable for publication. We look forward to your positive response and any further guidance you may have.

Thank you once again for your understanding and support. We are confident that our revised manuscript is now in a much better position to contribute to the field.

Sincerely,

Yanbin Zhu

Xizang Academy of Agriculture and Animal Husbandry Sciences, Lhasa
850000, China

E-mail address: zhuyanbin126@126.com

Re: Spectrum01001-25R1 (**Effects of warm-season feeding on yak growth, antioxidant capacity, immune function, and fecal microbiota**)

Dear Dr. yanbin Zhu:

Thank you for the privilege of reviewing your work. Below you will find my comments, instructions from the Spectrum editorial office, and the reviewer comments.

Revision Guidelines

Sincerely,
Jinshui Lin
Editor
Microbiology Spectrum

Reviewer #1 (Comments for the Author):

the authors had fully revised the whole manuscript, could be accepted to publish.

Reviewer #2 (Public repository details (Required)):

The accession number PRJNA1223345 did not return any results in the SRA database. Please recheck the number for errors and re upload the data if necessary.

Reviewer #2 (Comments for the Author):

The manuscript is well prepared and presented. However please recheck the NCBI accession number (PRJNA1223345), as it does not return any results in the SRA database. All other aspects of the submission appear to be in good order.

Review of Manuscript

Effects of warm-season feeding on yak growth, antioxidant capacity, immune function, and Fecal Microbiota

The manuscript is well-prepared, and clearly presented, and the study design and data analysis are appropriate. The authors have addressed the research questions effectively, and the conclusions are supported by the data.

However, I noticed that the NCBI accession number **PRJNA1223345** currently does not return any data in the SRA database. I kindly request the authors to verify and update the accession number to ensure the data is accessible for verification and reproducibility purposes.

Other than this, the manuscript is suitable for publication.

Dear Editor,

Thank you very much for your and the reviewers' valuable comments and suggestions. Based on your feedback, we have carefully revised the manuscript as follows:

Reviewer 2 left me a message suggesting that the authors further clarify the following points. Please respond as well.

1. Statistical Methods: A more detailed explanation of the statistical methods used, including how the sample size was determined, would strengthen the rigor of the analysis and ensure the robustness of the results.

Response: We have provided a more detailed explanation of the statistical methods used in our study. Specifically, we have elaborated on the sample size determination and the rationale behind our statistical approach. In the Methods section, we have clearly stated the number of replicates for each sample (lines 97, 106, and 18), which supports the robustness of our results. Additionally, we have included a detailed description of the statistical tests employed and the assumptions made to ensure the rigor of our analysis.

2. Treatment Clarification: It would be important to confirm whether the yaks received any treatments (e.g., antibiotics, anthelmintics, vaccines) during the trial, as these could potentially affect the gut microbiota results and overall interpretation of the findings.

Response: We have explicitly stated that none of the yaks received any antibiotics, vaccines, or other treatments during the trial (lines 109-110). This ensures that our findings on gut microbiota are not confounded by external treatments and provides a more accurate interpretation of our results.

3. Sequencing Data Quality: I recommend that the authors confirm the consistency and quality of the sequencing data, and include a more detailed description of sequencing depth, OTUs operational taxonomic units count, and detection limits in the methods section to enhance transparency.

Response: We have included a more detailed description of the sequencing data quality in the Methods section. Specifically, we have provided information on the sequencing depth, operational taxonomic units (OTUs) count, and detection limits. Additionally, we have also summarized the key details of the sequencing data in the Results section (lines 256-262) to ensure that readers can easily access this important information.

Reviewer #1

1. The author should carefully check the Latin terms in the manuscript, which should be italicized, such as "*Bos grunniens*" (line 45).

Response: We have carefully reviewed the manuscript and have italicized the Latin term "*Bos grunniens*" at the specified locations (lines 24 and 54). This change has been made to adhere to the standard scientific formatting conventions.

2. Consistency of Terminology: The author needs to standardize the use of specialized terms in the text. For example, expressions such as "penned fed," "Housing yak," and "pen-feeding" should be consistent to avoid confusion.

Response: We agree that standardizing specialized terms is crucial for clarity and coherence. In response to your comment, we have thoroughly reviewed the manuscript and standardized the use of terms related to the feeding and housing conditions of the yaks. Specifically, we have replaced

inconsistent expressions such as "penned fed," "Housing yak," and "pen-feeding" with the consistent term "housing yak" throughout the text.

3. At line 94, "yaks" should be in the singular form.

Response: In response to your comment at line 94, we have reviewed the usage of the term "yaks" and have made the necessary revisions to ensure grammatical accuracy.

We have changed "yaks" to the singular form "yak" at line 94, and we have also reviewed the entire manuscript to ensure consistent and correct usage of the singular and plural forms of "yak" throughout the text. This change aligns with the suggestions from both reviewers and enhances the clarity and coherence of our manuscript.

4. Microorganisms at the phylum level do not need to be italicized, such as "Actinobacteriota" at line 250 and "Bacteroidota" at line 251. The author should also carefully check the rest of the manuscript for consistency.

Response: We have carefully reviewed the manuscript and made the necessary corrections to ensure consistency and adherence to scientific conventions. In response to your comment, we have corrected the formatting of "Actinobacteriota" and "Bacteroidota" at line 267. These terms are now presented in regular typeface, as they should not be italicized at the phylum level. Additionally, we have conducted a thorough check of the entire manuscript to ensure that all phylum-level names are consistently formatted and not italicized.

5. At line 192, " $p < 0.05$ " should be capitalized as " $P < 0.05$ ".

Response: We have capitalized " p " to " P " at line 203, ensuring that it is consistent with the standard convention of capitalizing the " P " in " $P < 0.05$ " throughout the manuscript. This change enhances the accuracy and readability of our statistical reporting.

6. The author should refer to the journal's format and revise the citation style of the manuscript.

Response: We have updated the citations throughout the manuscript to conform to the journal's required format. The changes have been made in the reference section (lines 443-665) to ensure consistency and adherence to the journal's standards.

Reviewer #2 (Comments for the Author)

Reviewer #2 (Comments for the Author):

The study titled "Effects of Warm-Season Feeding Patterns on Growth Performance, Antioxidant Capacity, Immune Function, Metabolome, and Fecal Microbiota in Yaks" presents a well conducted and insightful analysis. It effectively integrates growth performance, immune function, metabolomics, and gut microbiota, making it highly relevant to yak husbandry and sustainable livestock production under climate change. However I suggest the following improvements to enhance clarity, scientific rigor, and readability follows as:

1. Title and Abstract: The title is informative but slightly lengthy. Consider simplifying it by grouping related terms or focusing on the key outcomes. The abstract is comprehensive but dense. Breaking up long sentences and reducing redundancy would enhance readability. Ensure *Bos grunniens* is italicized. Use the term "yaks" instead of "yak" for consistency and grammatical correctness.

Response:

Title, we have revised the title to make it more concise while retaining its informative nature. The new title is now more streamlined and focuses on the key outcomes of our study (revised in lines 1-4).

Abstract, We have revised the abstract to improve its readability. We have broken up long sentences and reduced redundancy to make the content more accessible. The revised abstract is now clearer and more concise (revised in lines 24-51).

Regarding the use of the term "yak" versus "yaks," we have reviewed the manuscript and have standardized the terminology for consistency and grammatical correctness. We have used "yak" in the singular form throughout the manuscript to maintain uniformity.

Additionally, we have ensured that "*Bos grunniens*" is italicized in the abstract and throughout the manuscript to adhere to scientific conventions.

2.Introduction: The introduction provides a well-rounded background on yak biology, ecological importance, and evolving husbandry practices. The rationale for comparing grazing and pen-feeding systems is presented. The hypothesis is logical but could be rephrased in a more concise and testable format. For example: "We hypothesize that warm-season grazing improves metabolic efficiency and immune responses compared to pen-feeding." Consider including one or two guiding research questions, such as: How do grazing and pen feeding differ in terms of long-term sustainability for yak populations, particularly under the pressure of climate change? Do shifts in gut microbiota in grazing yaks enhance metabolism or disease resistance?

Response: In response to your feedback, we have revised the Introduction to present the hypothesis in a more concise and testable format. We have also included guiding research questions to provide a clearer direction for our study. The revised hypothesis and research questions are now presented in lines 80-85

3.Materials and Methods. Please clarify whether the yaks received any antibiotic, anthelmintic, or vaccine treatments before or during the trial. This is critical for interpreting the microbiome and metabolomics data. Justify the selection of the V3-V4 region for 16S rRNA sequencing. Include basic sequencing statistics such as sequencing depth, OTUs count, and coverage to enhance transparency. The extraction and analysis methods are sound; however, including detection limits and rationale for the choice of solvents would increase methodological rigor.

Response: We have carefully considered your suggestions and have made the following revisions and clarifications:

Treatments for Yaks: We have explicitly stated in the manuscript that no antibiotics, anthelmintics, or vaccines were administered to the yaks before or during the trial. This information is clearly mentioned in the Methods section (lines 109-110) to ensure transparency and to support the interpretation of our microbiome and metabolomics data.

In addition, the V3-V4 region was chosen because it offers several advantages for microbial studies, including high species resolution, moderate sequence length, effective universal primer design, comprehensive database support, and good data comparability. We have included detailed sequencing statistics to enhance transparency. The sequencing depth, OTUs count, and coverage are described in both the Methods section (lines 158-170 under "Fecal microbiota") and the Results section (lines 252-258 under "Fecal

microbiota and short-chain fatty acids").

4. Results The results are structured. Including descriptive statistics (e.g., mean \pm SD) alongside p-values or significance indicators will improve interpretability. Try linking specific biomarker changes (e.g. immune cytokines or metabolites) to physiological processes such as liver function, nutrient metabolism, or muscle activity.

We have carefully considered your suggestions and have made the following revisions to enhance the clarity and interpretability of our results:

Descriptive Statistics and Significance Indicators: To improve the interpretability of our results, we have included descriptive statistics in the form of mean \pm SD alongside p-values or significance indicators. This information is now clearly presented in the tables, allowing readers to better understand the magnitude and significance of the observed changes.

Linking Biomarker Changes to Physiological Processes: We have strengthened the connection between specific biomarker changes (e.g., immune cytokines or metabolites) and physiological processes such as liver function, nutrient metabolism, or muscle activity. While the primary presentation of these links is in the Discussion section, we have also ensured that the results are framed in a way that highlights their relevance to these physiological processes.

5. Discussion: The discussion is comprehensive and well linked to the study objectives. Consider being more concise by focusing on the key findings and reducing redundancy, especially in sections about hormone regulation and immune responses. The integration of microbiota and metabolomics is a major strength this section would benefit from streamlining to highlight the novelty and significance of these findings.

We have carefully considered your suggestions and have revised the section to enhance its conciseness and focus on the key findings.

In response to your comments, we have streamlined the content related to hormone regulation and immune responses, reducing redundancy and focusing on the most significant results. This revision helps to highlight the novelty and importance of our findings, particularly in the integration of microbiota and metabolomics. The changes have been made in lines 304-351 to ensure that the discussion is more concise and impactful.

We believe these revisions improve the clarity and relevance of the Discussion section. Thank you again for your insightful comments.

6. Conclusion. The conclusion is generally satisfactory. It would be stronger if it briefly mentioned the underlying mechanisms and offered clear future research directions based on the study findings.

We have carefully considered your suggestions and have made the following revisions to enhance the strength and clarity of our conclusion.

In response to your comments, we have included a brief discussion of the underlying mechanisms observed in our study, as well as clear future research directions based on our findings. This additional information has been incorporated into the Conclusion section (lines 411-416).

We believe these changes provide a more comprehensive and forward-looking conclusion, highlighting the significance of our study and suggesting potential avenues for further research.

Reviewer #2 (Comments for the Author):

The study titled "Effects of Warm-Season Feeding Patterns on Growth Performance, Antioxidant Capacity, Immune Function, Metabolome, and Fecal Microbiota in Yaks" presents a well conducted and insightful analysis. It effectively integrates growth performance, immune function, metabolomics, and gut microbiota, making it highly relevant to yak husbandry and sustainable livestock production under climate change. However I suggest the following improvements to enhance clarity, scientific rigor, and readability follows as:

1. Title and Abstract: The title is informative but slightly lengthy. Consider simplifying it by grouping related terms or focusing on the key outcomes. The abstract is comprehensive but dense. Breaking up long sentences and reducing redundancy would enhance readability. Ensure *Bos grunniens* is italicized. Use the term "yaks" instead of "yak" for consistency and grammatical correctness.

Response:

Title, we have revised the title to make it more concise while retaining its informative nature. The new title is now more streamlined and focuses on the key outcomes of our study (revised in lines 1-4).

Abstract, we have revised the abstract to improve its readability. We have broken up long sentences and reduced redundancy to make the content more accessible. The revised abstract is now clearer and more concise (revised in lines 24-51).

Regarding the use of the term "yak" versus "yaks," we have reviewed the manuscript and have standardized the terminology for consistency and grammatical correctness. We have used "yak" in the singular form throughout the manuscript to maintain uniformity.

Additionally, we have ensured that "*Bos grunniens*" is italicized in the abstract and throughout the manuscript to adhere to scientific conventions.

2. Introduction: The introduction provides a well-rounded background on yak biology, ecological importance, and evolving husbandry practices. The rationale for comparing grazing and pen-feeding systems is presented. The hypothesis is logical but could be rephrased in a more concise and testable format. For example: "We hypothesize that warm-season grazing improves metabolic efficiency and immune responses compared to pen-feeding." Consider including one or two guiding research questions, such as: How do grazing and pen feeding differ in terms of long-term sustainability for yak populations, particularly under the pressure of climate change? Do shifts in gut microbiota in grazing yaks enhance metabolism or disease resistance?

Response: In response to your feedback, we have revised the Introduction to present the hypothesis in a more concise and testable format. We have also included guiding research questions to provide a clearer direction for our study. The revised hypothesis and research questions are now presented in lines 80-85

3. Materials and Methods. Please clarify whether the yaks received any antibiotic, anthelmintic, or vaccine treatments before or during the trial. This is critical for interpreting the microbiome and metabolomics data. Justify the selection of the V3-V4 region for 16S rRNA sequencing. Include basic sequencing statistics such as sequencing depth, OTUs count, and coverage to enhance transparency. The extraction and analysis methods are sound; however, including detection limits and rationale for the choice of solvents would increase methodological rigor.

Response: We have carefully considered your suggestions and have

made the following revisions and clarifications:

Treatments for Yaks: We have explicitly stated in the manuscript that no antibiotics, anthelmintics, or vaccines were administered to the yaks before or during the trial. This information is clearly mentioned in the Methods section (lines 109-110) to ensure transparency and to support the interpretation of our microbiome and metabolomics data.

In addition, the V3-V4 region was chosen because it offers several advantages for microbial studies, including high species resolution, moderate sequence length, effective universal primer design, comprehensive database support, and good data comparability. We have included detailed sequencing statistics to enhance transparency. The sequencing depth, OTUs count, and coverage are described in both the Methods section (lines 158-170 under "Fecal microbiota") and the Results section (lines 252-258 under "Fecal microbiota and short-chain fatty acids").

4.Results The results are structured. Including descriptive statistics (e.g., mean \pm SD) alongside p-values or significance indicators will improve interpret ability. Try linking specific biomarker changes (e.g immune cytokines or metabolites) to physiological processes such as liver function, nutrient metabolism, or muscle activity.

We have carefully considered your suggestions and have made the following revisions to enhance the clarity and interpretability of our results:

Descriptive Statistics and Significance Indicators: To improve the interpretability of our results, we have included descriptive statistics in the form of mean \pm SD alongside p-values or significance indicators. This information is now clearly presented in the tables, allowing readers to better understand the magnitude and significance of the observed changes.

Linking Biomarker Changes to Physiological Processes: We have strengthened the connection between specific biomarker changes (e.g., immune cytokines or metabolites) and physiological processes such as liver function, nutrient metabolism, or muscle activity. While the primary presentation of these links is in the Discussion section, we have also ensured that the results are framed in a way that highlights their relevance to these physiological processes.

5. Discussion: The discussion is comprehensive and well linked to the study objectives. Consider being more concise by focusing on the key findings and reducing redundancy, especially in sections about hormone regulation and immune responses. The integration of microbiota and metabolomics is a major strength this section would benefit from streamlining to highlight the novelty and significance of these findings.

We have carefully considered your suggestions and have revised the section to enhance its conciseness and focus on the key findings.

In response to your comments, we have streamlined the content related to hormone regulation and immune responses, reducing redundancy and focusing on the most significant results. This revision helps to highlight the novelty and importance of our findings, particularly in the integration of microbiota and metabolomics. The changes have been made in lines 304-351 to ensure that the discussion is more concise and impactful.

We believe these revisions improve the clarity and relevance of the Discussion section. Thank you again for your insightful comments.

6.Conclusion. The conclusion is generally satisfactory. It would be

stronger if it briefly mentioned the underlying mechanisms and offered clear future research directions based on the study findings.

We have carefully considered your suggestions and have made the following revisions to enhance the strength and clarity of our conclusion.

In response to your comments, we have included a brief discussion of the underlying mechanisms observed in our study, as well as clear future research directions based on our findings. This additional information has been incorporated into the Conclusion section (lines 411-416).

We believe these changes provide a more comprehensive and forward-looking conclusion, highlighting the significance of our study and suggesting potential avenues for further research.

7. Data availability: The data availability statement should be more specific. Please include an accession number (e.g. NCBI SRA) or provide a link to an online repository if available for better transparency.

We fully agree that providing specific details about the data repository enhances transparency and reproducibility.

In response to your suggestion, we have updated the Data Availability statement to include the accession number and a direct link to the NCBI SRA repository where our data are stored. This information can be found in lines 435-438.

Thank you very much for your time and effort in reviewing our manuscript. We truly appreciate the constructive comments and suggestions provided by you and the reviewers, which have been instrumental in helping us improve the quality of our work. In response to your feedback, we have made a series of revisions to enhance the clarity, accuracy, and overall presentation of our manuscript. We believe these changes have significantly strengthened our study and addressed the concerns raised.

We hope that these revisions meet your expectations and that you will find our manuscript suitable for publication. We look forward to your positive response and any further guidance you may have.

Thank you once again for your understanding and support. We are confident that our revised manuscript is now in a much better position to contribute to the field.

Sincerely,

Yanbin Zhu

Xizang Academy of Agriculture and Animal Husbandry Sciences, Lhasa
850000, China

E-mail address: zhuyanbin126@126.com

Re: Spectrum01001-25R2 (**Effects of warm-season feeding on yak growth, antioxidant capacity, immune function, and fecal microbiota**)

Dear Dr. yanbin Zhu:

Your manuscript has been accepted, and I am forwarding it to the ASM production staff for publication. Your paper will first be checked to make sure all elements meet the technical requirements. ASM staff will contact you if anything needs to be revised before copyediting and production can begin. Otherwise, you will be notified when your proofs are ready to be viewed.

Sincerely,
Jinshui Lin
Editor
Microbiology Spectrum